# Factors associated with domestic violence in the Lahu hill tribe of northern Thailand: A cross-sectional study

**Nicharuch Panjaphothiwat[1], Ratipark Tamornpark[1,2], Tawatchai Apidechkul●[1,2]\*, Prapamon Seeprasert[1], Onnalin Singkhorn[3], Panupong Upala[2], Phitnaree Thutsanti[4], Fartima Yeemard[1], Rachanee Sunsern[1]**

1 School of Health Science, Mae Fah Luang University, Chiang Rai, Thailand, 2 Center of Excellence for the Hill Tribe Health Research, Mae Fah Lung University, Chiang Rai, Thailand, 3 School of Nursing, Mae Fah Luang University, Chiang Rai, Thailand, 4 Department of Disease Control, Ministry of Public Health, Nonthaburi, Thailand

\* tawatchai.api@mfu.ac.th

**Data Availability Statement:** All relevant data are within the manuscript and its Supporting Information files.

## Abstract

### Background

Domestic violence significantly affects physical and mental health, particularly among children, women, and the elderly. Living in certain family environments could lead to victimization by domestic violence, especially among families with a poor socioeconomic status, such as the Lahu hill tribe people in Thailand. This study aimed to estimate the prevalence of and determine the factors associated with domestic violence among Lahu children, women, and the elderly.

### Methods

A cross-sectional study was conducted of participants who belonged to the Lahu hill tribe and lived in 20 selected villages in Chiang Rai Province, Thailand. A validated questionnaire was used to collect personal information and information regarding experiences related to domestic violence in the past year from children (aged 5–15 years), women (aged 16–59 years), and the elderly (aged 60 years and over). A binary logistic regression was used to detect associations between the variables.

### Results

A total of 646 participants were recruited into the study, specifically, 98 children aged 5–15 years, 430 women aged 16–59 years, and 118 elderly people. Children who smoked (AOR = 8.70; 95%CI = 1.27–59.45) had greater odds of experiencing domestic violence than children who did not smoke. Women who had a role as a family member (AOR = 1.59; 95%CI = 1.02–2.50), used alcohol (AOR = 3.36; 95%CI = 2.27–4.99), lived in a family with financial problems (AOR = 4.01; 95%CI = 2.52–7.66), and lived with a family member who uses alcohol (AOR = 2.87; 95%CI = 2.20–5.63) had greater odds of suffering domestic violence than women who did not share these characteristics. The elderly who used alcohol (AOR = 3.25,

**Funding:** The project was supported by The Center for Alcohol Studies, Thailand (Grant No 60-A-0011). Dr. Tawatchai Apidechkul was the principle investigator and received the grant. However, the grant funder did not involve in all steps of the study.

**Competing interests:** The authors declare that they have no competing interests.

95%CI = 1.08–9.81), lived with a family member who uses alcohol (AOR = 3.31; 95%CI = 1.26–7.34), or lived in a family with financial problems in the past year (AOR = 2.16; 95%CI = 1.16–8.77) had greater odds of facing domestic violence than the elderly who did not have these characteristics.

## Conclusion

Family financial problems and substance use are associated with domestic violence in Lahu families in Thailand. Health interventions to reduce the use of substances, including training programs to respond to domestic violence, should be promoted for Lahu children, women, and the elderly.

## Introduction

Domestic violence is widely recognized as a global concern and leads to public health and other social problems [1]. Children, women, and the elderly are the most vulnerable to domestic violence or intimate partner violence (IPV) [2]. There are four main types of violence, namely, physical violence, sexual violence, stalking, and psychological aggression [2]. The World Health Organization (WHO) has reported that more than one billion children aged 2–17 years have experienced physical, sexual, or emotional violence or neglect in the past year, which impacts lifelong health and well-being [3]. Moreover, target no. 16.2 of the Sustainable Development Goals (SDG) of the United Nations (UN) clearly states that all forms of violence against children, including abuse, exploitation, trafficking and torture, should be minimized in the next few years [4]. The WHO also confirms that violence against women, especially IPV and sexual violence, is a global problem [5]. It has been indicated that 35.0% of women worldwide have experienced either physical and/or sexual IPV at least once in their lives, which can eventually negatively affect women's physical, mental, sexual, and reproductive health [5]. One in six individuals among the elderly (16.6%) around the world has experienced some form of abuse from people in their community in the past year. Elder abuse leads to severe physical and psychological problems [6]. In 2019, the UN estimated that the total number of people aged 60 years and over will be more than 2 billion by 2050 [7]. They become vulnerable domestically, particularly the elderly who live in developing countries, including Thailand [1].

Thailand is a developing country and has been reported to have a high prevalence of domestic violence, particularly among children [8], women [9] and the elderly [10]. Most cases of domestic violence have been reported in populations that live in poor socioeconomic environments and have low levels of education [10–14]. The WHO has reported that children and women in Thailand living with people who use substances such as amphetamines and alcohol have a greater chance of being victimized by some form of domestic violence [15]. Alcohol use in Thai society is very common [16]. The integration between economic constraints and substance use has become a significant factor that contributes to domestic violence [17]. Adult men have been indicated as the contributors to domestic violence in Thai society who can make people in their families vulnerable to physical, mental, and sexual violence [18].

The hill tribes are a group of people who migrated from South China over the past two centuries and have settled in the northern region of Thailand [19]. There are six main groups, namely, the Akha, Lahu, Hmong, Yao, Karen, and Lisu, with approximately 3.5 million people of the hill tribes living in Thailand in 2019 [19]. The Lahu people are the second-largest hill tribe in Thailand [19, 20]. Lahu villages are settled along the border of Thailand and Myanmar

[21]. The Lahu people have their own language, lifestyle and perception with respect to using substances such as alcohol [22, 23] and amphetamines [24], particularly the young adult and adult populations. A large proportion of the Lahu people in Thailand live under the national poverty line, which is less than 2,500 baht (US$84) per month per person [25]. Under the current situation of the Lahu people in Thailand, information relevant to the prevalence of domestic violence against children, women, and the elderly, including information on the factors contributing to violence, is not available. This information is needed for use in public health policy development and implementation.

With several conditions related to the Lahu people, this study aimed to estimate the prevalence of and determine the factors associated with domestic violence, which occurs in children aged ≤ 15 years, women, and the elderly aged ≥60 years among the Lahu people in Thailand.

## Materials and methods

### Study design and settings

A cross-sectional study was conducted in 20 villages randomly selected out of 216 Lahu villages in Chiang Rai Province, Thailand [20], using a random number generation method. In the study process, only 20 Lahu villages were randomly selected from the list.

### Study populations

The study population comprised all Lahu people who live in the 20 selected villages. The targeted population was classified into the 3 groups: children aged 5–15 years, women aged 16–59 years, and older people aged 60 years and over. The entire targeted population lived in the selected villages at the date of data collection and met the inclusion criteria, while the people who could not provide essential information relevant to the study protocol were excluded from the study.

### Study sample and sample size calculation

The study sample size was calculated based on the formula for a cross-sectional study, that is, n = $[Z^2\alpha/_2{}^*P^*Q]/e^2$, where Z is the value from the standard normal distribution that corresponds to the desired confidence level (Z = 1.96 for 95% CI), P is the expected true proportion (which is considered based on the previous study), and e is the desired precision, which is 0.05. Based on a study conducted in Thailand in 2018, the prevalence of domestic violence was 15.0% [9]. Therefore, the minimum required sample size for the study was 196 participants for each subpopulation (children aged 5–15, women aged 16–59, and the elderly aged 60). When 10.0% was added to account for any possible errors in the study process, 646 individuals were required for the analyses.

### The research tool and its development

A questionnaire was developed for the study. The information used for questionnaire development was acquired from different sources, such as a literature review, discussions with psychologists who worked in the field, and discussions with Lahu village leaders. The validity and reliability of the questionnaire were improved before use. The validity of the questionnaire was detected and improved by an item-objective congruence (IOC) method that asked three external experts in the field to provide scores and comments. The questions that scored lower than 0.5 were deleted from the set of questionnaires, while the questions that scored 0.5–0.7 were considered to require revision before inclusion in the questionnaire. The questions that scored higher than 0.7 were included in the questionnaire.

The questionnaire was piloted with 20 Lahu people who lived in two villages in Mae Chan District, Chiang Rai Province, between February 1 and 20, 2019. The purpose was to improve the validation and reliability before using the questionnaire in the field. Moreover, it was also important to confirm the forms of domestic violence in Lahu people.

Ultimately, the questionnaire comprised five parts. In part one, 16 questions were used to collect data on the general characteristics of the participants, such as their age, sex, education and marital status. In part two, 7 questions were used to collect information on substance use, such as smoking, alcohol use, marijuana use, amphetamine use, glue use, opium use, and family members who used alcohol. In part three, 7 questions were used to collect information on women's experiences of harm related to sex, such as experiences with sexual harassment and being forced to have sex. In part four, 5 questions were used to collect information on domestic violence among children aged ≤15 years and the elderly aged ≥ 60 years, such as being left to live alone and being ignored regarding financial support and caregiving when they have had health problems. In part five, 12 questions were used to collect information on different kinds of domestic violence among children aged ≤ 15 years, women, and the elderly aged≥60 years, such as being forced to use substances, being insulted or addressed rudely and being addressed with anger to provoke a fearful reaction. The information from part 5 was used as the dependent variable in the study. The overall Cronbach's alpha [26] on the questions related to domestic violence among children, women, and the elderly were 0.70, 0.71, and 0.71. Domestic violence is defined as any form of violence or abuse that occurs in any relationship within a family [27]. In the study, the participants were asked about their experience of domestic violence through different questions, such as "have you experienced getting kicked out of the house by a family member in the past year?", "have you experienced being forced to drink alcohol, smoke, or use substances in the past year by a family member?" and "have you experienced being forced to ask for money or borrow items from others?".

## Steps of data collection

Access to all 20 selected villages was granted by district officers. Village headmen were contacted and provided with brief information regarding the study, and appointments were made to visit the village 5 days prior. At the time of data collection, the people who were willing to participate in the study were provided essential information and asked to provide an informed consent form before starting the interview. Participants aged less than 18 years and their parents were also asked to provide informed consent. For the children (aged 15 years and below), the information was obtained from their parents, except for the questions that asked about harms received. For these questions, the children were carefully questioned in a private room by a trained psychiatric nurse. People who were not able to provide essential information due to personal health problems were excluded from the study. Participants who could read Thai were asked to complete the questionnaire by themselves. However, for those who could not read Thai, one of the researchers who spoke Thai and Lahu conducted the interviews. The village health volunteers were asked for their help in obtaining information from the participants who could not communicate in Thai. The interviews were conducted in a private and confidential room that was provided in a community hall in the village and lasted for 30 minutes each. Data were collected from June-October 2019.

## Statistical analyses

Data were entered in duplicate into Excel sheets, and errors were checked before the data were transferred into SPSS version 24, 2020 (SPSS, Chicago, IL) for the analyses. Descriptive and inferential analyses were performed. Descriptive statistics were used to describe the

characteristics of the participants. Categorical data are presented as percentages, while continuous data with normal distributions are presented as the means and standard deviations (SDs). A binary logistic regression was used to detect the associations between the variables at a significance of $\alpha = 0.05$. The "Enter" mode was used in the step involving the selection of the independent variables into the statistical model. The pseudo $R^2$ of the Cox-Snell $R^2$ and Nagelkerke $R^2$ and the Hosmer-Lemshow chi-square were used to determine the fit of the model in all steps. In the univariable logistic model, all independent variables detected the association with the dependent variable one-by-one in each step. These variables found to be significant in the univariable logistic model were considered to be put into the multivariable model. Before fitting the final multivariable logistic model, some variables were controlled as the confounder factors before interpretation.

## Ethical considerations

The study's proposal and protocol were reviewed and approved by the Mae Fah Luang University Research Ethics Committee on Human Research (No. REH-60107). Before starting the interviews, all participants were provided all relevant and essential information. Informed consent was obtained on a voluntary basis before starting the interview. The interviews were closely monitored by a psychiatric nurse, and when they appeared to have a negative impact on the participants, it was recommended that the question be skipped or the interview be stopped. Afterward, all questionnaires were properly destroyed after coding and entering their information into an Excel spreadsheet. The data files were stored confidentially with a specific code for access.

## Results

A total of 646 participants were included in the analyses, namely, 98 children aged 5–15 years, 430 women aged 16–59 years, and 118 people aged 60 years and over. Among the children, 52.0% were male, and 70.4% were attending school. Among the women, 77.0% were married, 51.9% were Buddhist, 52.0% were illiterate, and 44.2% were farmers. Among the elderly, 53.4% were female, 68.5% were married, 61.2% had no income, and 26.3% had a medical illness (Table 1).

Different kinds of substance use were reported by the participants: 15.3% and 7.1% of children aged 5–15 years reported alcohol use and smoking, respectively. Moreover, 37.7% and 11.9% of the women and 15.3% and 49.2% of the elderly aged 60 years and over reported alcohol use and smoking, respectively (Table 2).

Six out of 98 children (6.1%) had experienced some form of domestic violence; 4.1% had been insulted or addressed rudely, 4.1% had been addressed with anger, 3.1% had been forced to ask for money from others, etc. Seventy-four women (17.2%) had experienced some form of domestic violence; 13.3% had experienced sexual abuse, 6.3% had been insulted or addressed rudely, 6.0% had been forced to buy food or drink, etc. Twenty-three of the 118 elderly (19.5%) had experienced some form of domestic violence; 4.2% had been forced to ask for money from others, 3.4% had been kicked out of the house, 3.4% had been forced to buy food or drink or forced to work, etc. (Table 3).

In the univariate analysis, smoking was found to be associated with experiencing domestic violence in Lahu children. In the multiple logistic regression analysis, children who smoked had 8.70 times (95%CI = 1.27–59.45) greater odds to suffer domestic violence than children who did not smoke (Table 4).

In the univariate analysis, it was found that four variables were associated with domestic violence in Lahu women, specifically, having family financial problems in the past year, having a role as a family member, using alcohol, and having a family member who uses alcohol.

**Table 1. Characteristics of the participants.**

| Characteristics | Total | Children aged 5–15 years | | Women aged 16–59 years | | Elderly aged ≥ 60 years | |
|---|---|---|---|---|---|---|---|
| | n (%) | n | % | n | % | n | % |
| **Total** | **646 (100.0)** | 98 | 100.0 | 430 | 100.0 | 118 | 100.0 |
| **Sex** | | | | | | | |
| Female | 540 (83.6) | 47 | 48.0 | 430 | 100.0 | 63 | 53.4 |
| Male | 106 (16.4) | 51 | 52.0 | N/A | N/A | 55 | 46.6 |
| **Marital status** | | | | | | | |
| Single | 157 (24.3) | 96 | 98.0 | 58 | 13.5 | 3 | 2.5 |
| Married | 414 (64.1) | 2 | 2.0 | 331 | 77.0 | 81 | 68.6 |
| Ever married | 75 (11.6) | 0 | 0.0 | 41 | 9.5 | 34 | 28.8 |
| **Religion** | | | | | | | |
| Buddhism | 348 (53.8) | 61 | 62.2 | 223 | 51.9 | 64 | 53.8 |
| Christianity | 298 (46.2) | 37 | 37.8 | 207 | 48.1 | 54 | 46.2 |
| **Education** | | | | | | | |
| Illiterate | 336 (52.0) | 4 | 4.1 | 217 | 50.5 | 115 | 97.5 |
| Primary school | 175 (27.0) | 68 | 69.4 | 105 | 24.4 | 2 | 1.7 |
| Secondary school and higher | 37 (21.0) | 26 | 26.5 | 108 | 25.1 | 1 | 0.8 |
| **Occupation** | | | | | | | |
| Student | 101 (15.6) | 69 | 70.4 | 28 | 6.5 | 4 | 3.4 |
| Unemployed | 122 (18.8) | 6 | 6.1 | 71 | 16.5 | 45 | 38.1 |
| Farmer | 253 (39.2) | 0 | 0.00 | 190 | 44.2 | 63 | 53.4 |
| Other | 170 (26.4) | 23 | 23.5 | 141 | 32.8 | 6 | 5.1 |
| **Having income** | | | | | | | |
| Yes | 391 (60.5) | 5 | 5.1 | 352 | 81.9 | 34 | 28.8 |
| No | 255 (39.5) | 93 | 94.9 | 78 | 18.1 | 84 | 71.2 |
| **Having family financial problems in the past year** | | | | | | | |
| Yes | 289 (46.1) | 22 | 22.4 | 201 | 46.7 | 66 | 55.9 |
| No | 348 (53.9) | 76 | 77.6 | 229 | 53.3 | 52 | 44.1 |
| **Medical illness** | | | | | | | |
| Yes | 71 (11.0) | 0 | 0.00 | 40 | 9.3 | 31 | 26.3 |
| No | 575 (89.0) | 98 | 100.0 | 390 | 90.7 | 87 | 73.7 |
| **Role in family** | | | | | | | |
| Head | 145 (22.4) | 0 | 0.00 | 76 | 17.7 | 69 | 58.5 |
| Member | 501 (77.6) | 98 | 100.0 | 354 | 82.3 | 49 | 41.5 |

In the multiple logistic regression analysis, four variables were found to be associated with domestic violence in Lahu women, namely, having a role as a family member, using alcohol, having family financial problems, and having a family member who uses alcohol. The women who had a role as a family member had 1.59 times (95%CI = 1.02–2.50) greater odds of experiencing domestic violence than the women who were the head of the family. The women who used alcohol had 3.36 times (95%CI = 2.27–4.99) greater odds of suffering domestic violence than the women who did not use alcohol. The women who had family financial problems in the past year had 4.01 times (95%CI = 2.52–7.66) greater odds of enduring domestic violence than the women who did not have family financial problems. The women who had a family member who used alcohol had 2.87 times (95%CI = 2.20–5.63) greater odds of experiencing domestic violence than the women who did not have a family member who used alcohol (Table 5).

Table 2. Substance use behaviors among Lahu children, women, and the elderly.

| Characteristic | Children aged 5–15 years | | Women aged 16–59 years | | Elderly aged $\geq$ 60 years | |
|---|---|---|---|---|---|---|
| | n | % | n | % | n | % |
| **Smoking** | | | | | | |
| Yes | 7 | 7.1 | 51 | 11.9 | 58 | 49.2 |
| No | 91 | 92.9 | 379 | 88.1 | 60 | 50.8 |
| **Alcohol use** | | | | | | |
| Yes | 15 | 15.3 | 162 | 37.7 | 15 | 15.3 |
| No | 83 | 84.7 | 268 | 62.3 | 83 | 84.7 |
| **Marijuana use** | | | | | | |
| Yes | 1 | 1.0 | 3 | 0.7 | 1 | 0.8 |
| No | 97 | 99.0 | 427 | 99.3 | 117 | 99.2 |
| **Amphetamine use** | | | | | | |
| Yes | 0 | 0.0 | 5 | 1.2 | 2 | 1.7 |
| No | 98 | 100.0 | 425 | 98.8 | 116 | 98.3 |
| **Glue use** | | | | | | |
| Yes | 0 | 0.0 | 2 | 0.5 | 0 | 0.0 |
| No | 98 | 100.0 | 428 | 99.5 | 118 | 100.0 |
| **Opium use** | | | | | | |
| Yes | 0 | 0.0 | 2 | 0.5 | 1 | 0.8 |
| No | 98 | 100.0 | 428 | 99.5 | 117 | 99.2 |
| **Family member who uses alcohol** | | | | | | |
| Yes | 28 | 28.6 | 136 | 31.6 | 36 | 30.5 |
| No | 70 | 71.4 | 294 | 68.4 | 82 | 69.5 |

In the univariate analyses, the following four variables were found to be associated with domestic violence in the Lahu elderly: having family financial problems in the past year; smoking; using alcohol; and having a family member who uses alcohol.

In the multiple logistic regression analysis, three variables were found to be associated with domestic violence in the Lahu elderly as follows: having family financial problems in the past year; using alcohol; and having a family member who uses alcohol. The elderly who used alcohol had 3.25 times (95%CI = 1.08–9.81) greater odds of suffering domestic violence than the elderly who did not use alcohol. The elderly who had a family member who uses alcohol had 3.31 times (95%CI = 1.26–7.34) greater odds of experiencing domestic violence than the elderly who did not have a family member who uses alcohol. The elderly who had family financial problems in the past year had 2.16 times (95%CI = 1.16–8.77) greater odds of undergoing domestic violence than the elderly who did not have family financial problems (95%CI = 1.16–8.77) (Table 6).

## Discussion

The Lahu people in Thailand have a poor socioeconomic status and work as agriculturalists. With respect to substances, using alcohol and smoking are the most common practices. Children, women, and the elderly are vulnerable to domestic violence, and one-fifth of them had experienced domestic violence in the past year. A wide range of physical and mental abuse was reported, from low to heavy levels of harm, such as being asked to kill the vulnerable individuals in Lahu families. Some characteristics were detected as the factors associated with the presence of domestic violence in Lahu families, such as using alcohol, having a family member who uses alcohol, having family financial problems in the past year, and smoking.

**Table 3. Characteristics of domestic violence among Lahu children, women, and the elderly in the past year.**

| Characteristic | Children aged 5–15 years | | Women aged 16–59 years | | Elderly aged ≥ 60 years | |
|---|---|---|---|---|---|---|
| | **n** | **%** | **n** | **%** | **n** | **%** |
| **Total participants** | **98** | **100.0** | **430** | **100.0** | **118** | **100.0** |
| **Total experiencing domestic violence (persons)** | | | | | | |
| Yes | 6 | 6.1 | 74 | 17.2 | 23 | 19.5 |
| No | 92 | 93.9 | 356 | 82.8 | 95 | 80.5 |
| *Physical abuse* | | | | | | |
| Getting kicked out of the house | | | | | | |
| Yes | 0 | 0.0 | 10 | 2.3 | 4 | 3.4 |
| No | 98 | 100.0 | 420 | 97.7 | 114 | 96.6 |
| Being pushed, pulled or scratched or having items thrown at them | | | | | | |
| Yes | 0 | 0.0 | 10 | 2.3 | 2 | 1.7 |
| No | 98 | 100.0 | 420 | 97.7 | 116 | 98.3 |
| Being slapped, hit, kicked or strangled | | | | | | |
| Yes | 0 | 0.0 | 4 | 0.9 | 2 | 1.7 |
| No | 98 | 100.0 | 426 | 99.1 | 116 | 98.3 |
| Being threatened with a weapon | | | | | | |
| Yes | 0 | 0.0 | 2 | 0.5 | 0 | 0 |
| No | 98 | 100.0 | 428 | 99.5 | 118 | 100.0 |
| Being forced to drink alcohol, smoke, or use substances | | | | | | |
| Yes | 2 | 2.0 | 19 | 4.4 | 2 | 1.7 |
| No | 96 | 98.0 | 411 | 95.6 | 116 | 98.3 |
| *Mental abuse* | | | | | | |
| Being insulted or addressed rudely | | | | | | |
| Yes | 4 | 4.1 | 27 | 6.3 | 2 | 1.7 |
| No | 94 | 95.9 | 403 | 93.7 | 116 | 98.3 |
| Being addressed with anger | | | | | | |
| Yes | 4 | 4.1 | 25 | 5.8 | 2 | 1.7 |
| No | 94 | 95.9 | 405 | 94.2 | 116 | 98.3 |
| Being told that another person in the household will kill himself or herself | | | | | | |
| Yes | 0 | 0.0 | 5 | 1.2 | 3 | 2.5 |
| No | 98 | 100.0 | 425 | 98.8 | 115 | 97.5 |
| Being asked to kill others | | | | | | |
| Yes | 1 | 1.0 | 2 | 0.5 | 3 | 2.5 |
| No | 97 | 99.0 | 428 | 99.5 | 115 | 97.5 |
| Being forced to buy food or drink or being forced to work | | | | | | |
| Yes | 2 | 2.0 | 26 | 6.0 | 4 | 3.4 |
| No | 96 | 98.0 | 404 | 94.0 | 114 | 96.6 |
| Being forced to ask for money or borrow items from others | | | | | | |
| Yes | 3 | 3.1 | 22 | 5.1 | 5 | 4.2 |
| No | 95 | 96.9 | 408 | 94.9 | 113 | 95.8 |
| *Sexual abuse* ** *(only women)* | | | | | | |
| Yes | 1 | 2.0 | 57 | 13.3 | 1 | 1.6 |
| No | 48 | 98.0 | 373 | 86.7 | 62 | 98.4 |

In the Lahu family context, it was found that the prevalence of domestic violence in the three major victim groups of children (aged 5–15 years), women (aged 16–59 years), and the elderly (aged ≥ 60 years) in the past year was 6.1%, 17.2%, and 19.5%, respectively. The WHO

**Table 4. Univariate and multivariate analyses of the factors associated with domestic violence among Lahu children.**

| Characteristic | Domestic violence | | | | OR | 95%CI | p-value | AOR | 95%CI | p-value |
|---|---|---|---|---|---|---|---|---|---|---|
| | Yes | | No | | | | | | | |
| | n | % | n | % | | | | | | |
| Total | **6** | **6.1** | **92** | **93.9** | NA | NA | NA | NA | NA | NA |
| **Age** (years) | | | | | | | | | | |
| 5–10 | 1 | 2.6 | 37 | 97.4 | 1.00 | | | | | |
| 11–15 | 5 | 8.3 | 55 | 91.7 | 3.36 | 0.38–29.97 | 0.227 | | | |
| **Marital status** | | | | | | | | | | |
| Single | 6 | 6.3 | 90 | 93.8 | NA | | | | | |
| Married | 0 | 0.0 | 2 | 100.0 | NA | NA | NA | | | |
| **Religion** | | | | | | | | | | |
| Buddhism | 2 | 3.3 | 59 | 96.7 | 1.00 | | | | | |
| Christianity | 4 | 10.8 | 33 | 89.2 | 3.58 | 0.62–20.58 | 0.154 | | | |
| **Education** | | | | | | | | | | |
| Illiterate | 1 | 25.0 | 3 | 75.0 | 2.56 | 0.20–33.16 | 0.473 | | | |
| Primary school | 2 | 2.9 | 66 | 97.1 | 0.23 | 0.04–1.48 | 0.122 | | | |
| Secondary school and higher | 3 | 11.5 | 23 | 88.5 | 1.00 | | | | | |
| **Occupation** | | | | | | | | | | |
| Student | 4 | 5.8 | 65 | 94.2 | 1.00 | | | | | |
| Employed and unemployed | 2 | 6.9 | 27 | 93.1 | 1.20 | 0.20–6.96 | 0.836 | | | |
| **Having income** | | | | | | | | | | |
| No | 5 | 5.4 | 88 | 94.6 | 1.00 | | | | | |
| Yes | 1 | 20.0 | 4 | 80.0 | 4.40 | 0.41–47.04 | 0.220 | | | |
| **Having family financial problems in the past year** | | | | | | | | | | |
| No | 2 | 9.1 | 20 | 90.9 | 1.00 | | | | | |
| Yes | 4 | 5.3 | 72 | 94.7 | 0.55 | 0.09–3.05 | 0.514 | | | |
| **Health illness** | | | | | | | | | | |
| No | 6 | 6.1 | 92 | 93.9 | NA | | | | | |
| Yes | 0 | 0.0 | 0 | 0.0 | NA | NA | NA | | | |
| **Role in family** | | | | | | | | | | |
| Head | 6 | 6.1 | 92 | 93.9 | NA | | | | | |
| Member | 0 | 0.0 | 0 | 0.0 | NA | NA | NA | | | |
| **Smoking** | | | | | | | | | | |
| No | 4 | 4.4 | 87 | 95.6 | 1.00 | | | 1.00 | | |
| Yes | 2 | 28.6 | 5 | 71.4 | 8.70 | 1.27–59.45 | 0.027* | 8.70 | 1.27–59.45 | 0.027* |
| **Alcohol use** | | | | | | | | | | |
| No | 4 | 4.8 | 79 | 95.2 | 1.00 | | | | | |
| Yes | 2 | 13.3 | 13 | 86.7 | 3.04 | 0.51–18.31 | 0.225 | | | |
| **Having a family member who uses alcohol** | | | | | | | | | | |
| Yes | 3 | 10.7 | 25 | 89.3 | 2.68 | 0.50–14.16 | 0.245 | | | |
| No | 3 | 4.3 | 67 | 95.7 | 1.00 | | | | | |
| **Marijuana use** | | | | | | | | | | |
| No | 6 | 6.2 | 91 | 93.8 | NA | | | | | |
| Yes | 0 | 0.0 | 1 | 100.0 | NA | NA | NA | | | |
| **Amphetamine use** | | | | | | | | | | |
| No | 6 | 6.1 | 92 | 93.9 | NA | | | | | |
| Yes | 0 | 0.0 | 0 | 0.0 | NA | NA | NA | | | |
| **Glue use** | | | | | | | | | | |

*(Continued)*

**Table 4.** (Continued)

| Characteristic | Domestic violence | | | | OR | 95%CI | p-value | AOR | 95%CI | p-value |
|---|---|---|---|---|---|---|---|---|---|---|
| | Yes | | No | | | | | | | |
| | n | % | n | % | | | | | | |
| No | 6 | 6.1 | 92 | 93.9 | NA | | | | | |
| Yes | 0 | 0.0 | 0 | 0.0 | NA | NA | NA | | | |
| **Opium use** | | | | | | | | | | |
| No | 6 | 6.1 | 92 | 93.9 | NA | | | | | |
| Yes | 0 | 0.0 | 0 | 0.0 | NA | NA | NA | | | |

* Significance level at α = 0.05

** Significance level at α = 0.05 after controlling for religion and education.

[28] has reported that the rate of domestic violence against women is 30.0%, which is higher than the estimates in the Lahu community in Thailand. A very interesting meta-analysis of the prevalence and health outcomes of domestic violence among the people attended to in hospitals of Arab countries found that 73.3% of women had been exposed to some form of domestic violence in their lifetimes [29]. Regarding domestic violence against children, the Centers for Disease Control and Prevention (CDC) reported that 11.0% of children have been exposed to some form of family violence in the past year in American families [30], while a prevalence of 34.0% was reported in Canada [31]. The WHO has also presented that the problem of child abuse is related to family economics, parents' education, parenting styles, and other environmental factors [28]. Regarding the elderly, a systematic review study reported that the prevalence of domestic violence for the elderly (aged ≥ 60 years) was 5.6–14.1%, including neglecting family members' essential needs for daily life in the past year [32]. Thailand is reported to have a 14.0% prevalence of domestic violence among the elderly [10]. This problem increases according to family economics and chronic health problems in the elderly [10]. Accordingly, Lahu children and women are living with less domestic violence than other communities, while the Lahu elderly have a greater chance of experiencing domestic violence than the elderly in other communities.

Among Lahu children, it was found that smoking is associated with them suffering domestic violence. This finding is supported by a study in Thailand [33], which reported that any form of family member smoking behavior is related to experiencing domestic violence. Sharma, et al. [34] clearly demonstrated the relationship between family member smoking and family financial problems. Moreover, smoking and amphetamine use are closely related, particularly among the Lahu people, which is supported by Apidechkul et al. [24]. Parents do not desire to see their child involved in amphetamine use, which is illegal in Thailand, and this results in domestic violence toward children. Many studies have reported and confirmed the association between substance use and domestic violence in both straight and reverse effects [15, 23, 24, 34, 35]. Moreover, on explanation is that parents do not want to see their child smoking; therefore, domestic violence could be occurring to stop children from smoking [23].

Lahu women who are defined as a family member with a role in in their family are at a greater risk of experiencing domestic violence than Lahu women who are the head of the family. This is supported by a study in Jordan that reported that women who lived with an extended family and who were not the family head were at risk of suffering domestic violence more than women who were the family head [36]. Moreover, the WHO has reported that women in many societies and families are commonly identified as one of the key victims of domestic violence due to a lack of power to protect themselves from their husband due to their

**Table 5. Univariate and multivariate analyses of the factors associated with domestic violence among Lahu women.**

| Characteristic | Domestic violence | | | | OR | 95%CI | p-value | AOR | 95%CI | p-value |
|---|---|---|---|---|---|---|---|---|---|---|
| | Yes | | No | | | | | | | |
| | n | % | n | % | | | | | | |
| Total | 74 | 17.2 | 356 | 82.8 | NA | NA | NA | NA | NA | NA |
| **Age** (years) | | | | | | | | | | |
| 16–30 | 35 | 20.0 | 140 | 80.0 | 1.56 | 0.81–3.03 | 0.182 | | | |
| 31–45 | 24 | 16.4 | 122 | 83.6 | 1.23 | 0.61–2.48 | 0.557 | | | |
| 46–59 | 15 | 13.8 | 94 | 86.2 | 1.00 | | | | | |
| **Marital status** | | | | | | | | | | |
| Single | 8 | 13.8 | 50 | 86.2 | 1.00 | | | | | |
| Married | 64 | 19.3 | 267 | 80.7 | 1.50 | 0.68–3.32 | 0.319 | | | |
| Ever married | 2 | 4.9 | 39 | 95.1 | 0.32 | 0.06–1.60 | 0.165 | | | |
| **Religion** | | | | | | | | | | |
| Buddhism | 34 | 15.2 | 189 | 84.8 | 1.00 | | | | | |
| Christianity | 40 | 19.3 | 167 | 80.7 | 1.33 | 0.81–2.20 | 0.264 | | | |
| **Education** | | | | | | | | | | |
| Illiterate | 35 | 16.1 | 182 | 83.9 | 1.11 | 0.58–2.10 | 0.759 | | | |
| Primary school | 23 | 21.9 | 82 | 78.1 | 1.61 | 0.80–3.26 | 0.183 | | | |
| Secondary school and higher | 16 | 14.8 | 92 | 85.2 | 1.00 | | | | | |
| **Occupation** | | | | | | | | | | |
| Student | 2 | 7.1 | 26 | 92.9 | 1.00 | | | | | |
| Unemployed | 17 | 23.9 | 54 | 76.1 | 4.09 | 0.88–19.05 | 0.073 | | | |
| Farmer | 37 | 19.5 | 153 | 80.5 | 3.14 | 0.71–13.84 | 0.130 | | | |
| Employed | 18 | 12.8 | 123 | 87.2 | 1.90 | 0.42–8.71 | 0.407 | | | |
| **Having income** | | | | | | | | | | |
| No | 16 | 20.5 | 62 | 79.5 | 1.31 | 0.71–2.43 | 0.394 | | | |
| Yes | 58 | 16.5 | 294 | 83.5 | 1.00 | | | | | |
| **Having family financial problems in the past year** | | | | | | | | | | |
| No | 18 | 62.1 | 211 | 37.9 | 1.00 | | | | | |
| Yes | 56 | 27.9 | 145 | 72.1 | 4.52 | 2.55–8.01 | <0.001* | 4.01 | 2.52–7.66 | <0.001* |
| **Health illness** | | | | | | | | | | |
| No | 68 | 17.4 | 322 | 82.6 | 1.00 | | | | | |
| Yes | 6 | 15.0 | 34 | 85.0 | 0.83 | 0.33–2.06 | 0.697 | | | |
| **Role in family** | | | | | | | | | | |
| Head | 7 | 9.2 | 69 | 90.8 | 1.00 | | | | | |
| Member | 67 | 18.9 | 287 | 81.1 | 2.30 | 1.01–5.23 | 0.047* | 1.59 | 1.02–2.50 | 0.042* |
| **Smoking** | | | | | | | | | | |
| No | 65 | 17.2 | 314 | 82.8 | 1.00 | | | | | |
| Yes | 9 | 17.6 | 42 | 82.4 | 1.04 | 0.48–2.23 | 0.930 | | | |
| **Alcohol use** | | | | | | | | | | |
| No | 36 | 13.4 | 232 | 86.6 | 1.00 | | | | | |
| Yes | 38 | 23.5 | 124 | 76.5 | 1.98 | 1.19–3.27 | 0.008* | 3.36 | 2.27–4.99 | <0.001* |
| **Having a family member who uses alcohol** | | | | | | | | | | |
| Yes | 43 | 31.6 | 93 | 68.4 | 3.92 | 2.33–6.59 | <0.001* | 2.87 | 2.20–5.63 | <0.001* |
| No | 31 | 10.5 | 263 | 89.5 | 1.00 | | | | | |
| **Marijuana use** | | | | | | | | | | |
| No | 72 | 16.9 | 355 | 83.1 | 1.00 | | | | | |
| Yes | 2 | 66.7 | 1 | 33.3 | 9.86 | 0.88–110.21 | 0.063 | | | |

*(Continued)*

**Table 5.** (Continued)

| Characteristic | Domestic violence | | | | OR | 95%CI | p-value | AOR | 95%CI | p-value |
|---|---|---|---|---|---|---|---|---|---|---|
| | Yes | | No | | | | | | | |
| | n | % | n | % | | | | | | |
| **Amphetamine use** | | | | | | | | | | |
| No | 72 | 16.9 | 353 | 83.1 | 1.00 | | | | | |
| Yes | 2 | 40.0 | 3 | 60.0 | 3.27 | 0.54–19.91 | 0.199 | | | |
| **Glue use** | | | | | | | | | | |
| No | 73 | 17.1 | 355 | 82.9 | 1.00 | | | | | |
| Yes | 1 | 50.0 | 1 | 50.0 | 4.86 | 0.30–78.64 | 0.265 | | | |
| **Opium use** | | | | | | | | | | |
| No | 74 | 17.3 | 354 | 82.7 | NA | | | | | |
| Yes | 0 | 0.0 | 2 | 100.0 | NA | NA | NA | | | |

* Significance level at α = 0.05

** Significance level at α = 0.05 after controlling for age, religion and education.

role in their family [28]. With less power in the family, Lahu women are disadvantaged by any family decision, and when they disagree with their husband's decision, violence can occur.

Interestingly, having financial problems in the past year in a Lahu family was found as one of the factors associated with domestic violence against women aged 16–59 years and against the elderly aged 60 years and over. This is supported by a study conducted by Stylianou [37]. Another demonstration of the relationship between family financial problems and domestic violence was reported by the Queensland Domestic and Family Violence Research Centre in 2017 [38]. The WHO has also reported that family financial problems are one of the contributors to domestic violence, particularly toward women and the elderly [39]. Additionally, in this study, it was found that the Lahu people in Thailand have a poor family economic status. This poor family economic status can easily contribute to domestic violence against women and the elderly.

In our study, it was found that Lahu women and the elderly were more likely to experience domestic violence while living with family members who use alcohol. This is confirmed by an official report by the WHO [40] that showed that alcohol is related to the occurrence of domestic violence in any society. A study in the United Kingdom (UK) also showed that alcohol was detected as a major cause of domestic violence [41]. Chan [42] and Yates [43] clearly demonstrated the association between alcohol use and domestic violence in the Australian context. In Thailand, a retrospective study conducted by Tongsamsi [16] reported that alcohol was found to be the major cause of domestic violence in Thailand. Moreover, from the national survey of smoking and alcohol in Thailand in 2017, it was found that families with members who use alcohol show significantly greater evidence of domestic violence than families with members who do not use alcohol [44]. According to the nature of the Lahu people, particularly young adults, they are very familiar with alcohol use [23], and after drinking alcohol, it would be easy for the elderly in the family to experience violence in some form.

Some participants could not understand the questions well, and the community health volunteers were asked to help as translators to improve the understanding of the context of the question asked before the response. Given the methodology used, the responses to some sensitive questions may not have been accurate, such as the pattern of experiencing some type of harm and amphetamine use. As per their norms, the Lahu people do not typically tell family stories to the public, particularly to individuals of a different gender. To minimize this

**Table 6. Univariate and multivariate analyses of the factors associated with domestic violence among the Lahu elderly.**

| Characteristic | Domestic violence | | | | OR | 95%CI | p-value | AOR | 95%CI | p-value |
|---|---|---|---|---|---|---|---|---|---|---|
| | Yes | | No | | | | | | | |
| | n | % | n | % | | | | | | |
| Total | 23 | 19.5 | 95 | 80.5 | NA | NA | NA | NA | NA | NA |
| **Age** (years) | | | | | | | | | | |
| 60–70 | 19 | 21.8 | 68 | 78.2 | 1.88 | 0.59–6.06 | 0.287 | | | |
| >70 | 4 | 12.9 | 27 | 87.1 | 1.00 | | | | | |
| **Marital status** | | | | | | | | | | |
| Single | 1 | 33.3 | 2 | 66.7 | 2.33 | 0.18–30.10 | 0.516 | | | |
| Married | 16 | 19.8 | 65 | 80.2 | 1.15 | 0.41–3.24 | 0.793 | | | |
| Ever married | 6 | 17.6 | 28 | 82.4 | 1.00 | | | | | |
| **Religion** | | | | | | | | | | |
| Buddhism | 9 | 14.1 | 55 | 85.9 | 1.00 | | | | | |
| Christianity | 14 | 25.9 | 40 | 74.1 | 2.14 | 0.84–5.43 | 0.110 | | | |
| **Attended school** | | | | | | | | | | |
| No | 21 | 18.3 | 94 | 81.7 | 1.00 | | | | | |
| Yes | 2 | 66.7 | 1 | 33.3 | 8.95 | 0.78–103.39 | 0.079 | | | |
| **Having income** | | | | | | | | | | |
| No | 3 | 8.8 | 31 | 91.2 | 3.23 | 0.89–11.70 | 0.074 | | | |
| Yes | 20 | 23.8 | 64 | 76.2 | 1.00 | | | | | |
| **Having family financial problems in the past year** | | | | | | | | | | |
| No | 5 | 9.6 | 47 | 90.4 | 1.00 | | | | | |
| Yes | 18 | 27.3 | 48 | 72.7 | 3.50 | 1.20–10.27 | 0.020* | 2.61 | 1.16–8.77 | 0.018* |
| **Health illness** | | | | | | | | | | |
| No | 19 | 21.8 | 68 | 78.2 | 1.00 | | | | | |
| Yes | 4 | 12.9 | 27 | 87.1 | 0.53 | 0.16–1.70 | 0.287 | | | |
| **Role in family** | | | | | | | | | | |
| Head | 15 | 78.3 | 54 | 78.3 | 1.42 | 0.55–3.68 | 0.466 | | | |
| Member | 8 | 16.3 | 41 | 83.7 | 1.00 | | | | | |
| **Smoking** | | | | | | | | | | |
| No | 5 | 8.3 | 55 | 91.7 | 1.00 | | | | | |
| Yes | 18 | 31.0 | 40 | 69.0 | 4.95 | 1.70–14.45 | 0.003* | | | |
| **Alcohol use** | | | | | | | | | | |
| No | 9 | 11.3 | 71 | 88.8 | 1.00 | | | 1.00 | | |
| Yes | 14 | 36.8 | 24 | 63.2 | 4.60 | 1.77–11.98 | 0.002* | 3.25 | 1.08–9.81 | 0.036* |
| **Having a family member who uses alcohol** | | | | | | | | | | |
| Yes | 13 | 36.1 | 23 | 63.9 | 4.06 | 1.57–10.50 | 0.003* | 3.31 | 1.26–7.34 | 0.001* |
| No | 10 | 12.2 | 72 | 87.8 | 1.00 | | | | | |
| **Marijuana use** | | | | | | | | | | |
| No | 23 | 19.7 | 94 | 80.3 | NA | | | | | |
| Yes | 0 | 0.0 | 1 | 100.0 | NA | NA | NA | | | |
| **Amphetamine use** | | | | | | | | | | |
| No | 23 | 19.8 | 93 | 80.2 | NA | | | | | |
| Yes | 0 | 0.0 | 2 | 100.0 | NA | NA | NA | | | |
| **Opium use** | | | | | | | | | | |
| No | 22 | 18.8 | 95 | 81.2 | NA | | | | | |

(*Continued*)

**Table 6.** (Continued)

| Characteristic | Domestic violence | | | | OR | 95%CI | p-value | AOR | 95%CI | p-value |
|---|---|---|---|---|---|---|---|---|---|---|
| | Yes | | No | | | | | | | |
| | n | % | n | % | | | | | | |
| Yes | 1 | 100.0 | 0 | 0.0 | NA | NA | NA | | | |

\* Significance level at α = 0.05

\*\* Significance level at α = 0.05 after controlling for age, religion and education.

problem, we used gender-matched interviewers and participants to collect the data. Moreover, as per the Lahu culture, the participants were not familiar with talking about domestic conflicts or conflicts within the family with people outside the family.

## Conclusion

The Lahu people in Thailand live with a low education level and work as agriculturists. Having three generations (grandparents, parents, and children) live together in the Lahu family is common. Smoking in children leads to them experiencing domestic violence, while alcohol use and family financial problems lead to the elderly suffering domestic violence. Regarding Lahu women experiencing domestic violence, using alcohol, having a family member who uses alcohol, having a role as a family member in their family and family financial problems are detected as the contributing factors. For recommendations to stop domestic violence at the individual level, the Lahu people should be encouraged to reduce or stop alcohol consumption and smoking. Improving personal skills to detect and prevent violence perpetrated by family members is also important. At the community level, all stakeholders in a community, including community leaders and health workers, have to be trained in family violence detection and reduction, particularly in conducting a campaign to minimize substance use among villagers. At the policy maker level, there should be concern, and policies should be formed to reduce the substance use problem and improve working skills for the Lahu people.

## Supporting information

**S1 Appendix. Questionnaire used in the study (Thai version).**
(PDF)

**S2 Appendix. Questionnaire used in the study (English version).**
(PDF)

**S3 Appendix. Data file of the study.**
(SAV)

## Acknowledgments

We would like to thank all the village headmen and participants for providing all the essential information.

## Author Contributions

**Conceptualization:** Nicharuch Panjaphothiwat, Tawatchai Apidechkul, Rachanee Sunsern.

**Data curation:** Nicharuch Panjaphothiwat, Ratipark Tamornpark, Tawatchai Apidechkul, Prapamon Seeprasert, Fartima Yeemard.

**Formal analysis:** Nicharuch Panjaphothiwat, Tawatchai Apidechkul.

**Funding acquisition:** Tawatchai Apidechkul.

**Investigation:** Ratipark Tamornpark, Tawatchai Apidechkul, Prapamon Seeprasert, Onnalin Singkhorn, Panupong Upala, Phitnaree Thutsanti, Fartima Yeemard, Rachanee Sunsern.

**Methodology:** Ratipark Tamornpark, Onnalin Singkhorn, Panupong Upala, Phitnaree Thutsanti.

**Supervision:** Rachanee Sunsern.

**Writing – original draft:** Nicharuch Panjaphothiwat, Ratipark Tamornpark, Tawatchai Apidechkul, Prapamon Seeprasert, Onnalin Singkhorn, Panupong Upala, Phitnaree Thutsanti, Fartima Yeemard.

**Writing – review & editing:** Nicharuch Panjaphothiwat, Ratipark Tamornpark, Tawatchai Apidechkul, Prapamon Seeprasert, Onnalin Singkhorn, Panupong Upala, Phitnaree Thutsanti, Fartima Yeemard, Rachanee Sunsern.

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
