## [Decision Letter · Decision Letter 0]

29 Oct 2020

PONE-D-20-27497

Factors associated with domestic violence in the Lahu hill tribe of northern Thailand: A cross-sectional study

PLOS ONE

Dear Dr. Apidechkul,

Thank you for submitting your manuscript to PLOS ONE. After careful consideration, we feel that it has merit but does not fully meet PLOS ONE’s publication criteria as it currently stands. Therefore, we invite you to submit a revised version of the manuscript that addresses the points raised during the review process.

We look forward to receiving your revised manuscript.

Kind regards,

Siyan Yi, MD, MHSc, PhD

Academic Editor

PLOS ONE

Additional Editor Comments:

We have received comments from three reviewers. One reviewer recommended rejection, one recommended minor revisions, and the other one recommended major revisions. The primary concerns are around the sampling and data analysis methods. I agree with two reviewers that stratified analyses should have been performed to explore the differences in domestic violence prevalence and its risk factors in different sub-populations. The sub-populations included girls and boys under 15 years, women and men, and female and male elderly over 60 years. The authors expressed an interest in studying the subgroups and should have reflected this in the analyses. The lack of sub-group results introduced a challenge in discussing and comparing this study's findings against other studies. We will consider this manuscript for further review if the authors could address these concerns.

Journal Requirements:

3. Please include a copy of the questionnaire, in the original language,as Supporting Information.

4. In statistical methods, please clarify whether you corrected for multiple comparisons.

5. Cross-sectional study design cannot be used to determine causation. In this light, please revise your conclusions to indicate that your study presents associations between exposure variables and domestic violence. For example, your statement that "smoking and alcohol and amphetamine use, leads to domestic violence" implies causation which is cannot be determined using your study design.

6. Please amend your list of authors on the manuscript to ensure that each author is linked to an affiliation. Authors’ affiliations should reflect the institution where the work was done (if authors moved subsequently, you can also list the new affiliation stating “current affiliation:….” as necessary).

7. Your ethics statement should only appear in the Methods section of your manuscript. If your ethics statement is written in any section besides the Methods, please move it to the Methods section and delete it from any other section. Please ensure that your ethics statement is included in your manuscript, as the ethics statement entered into the online submission form will not be published alongside your manuscript.

Reviewers' comments:

Reviewer's Responses to Questions

**Comments to the Author**

1. Is the manuscript technically sound, and do the data support the conclusions?

Reviewer #1: No

Reviewer #2: Partly

Reviewer #3: Partly

2. Has the statistical analysis been performed appropriately and rigorously? 

Reviewer #1: No

Reviewer #2: Yes

Reviewer #3: No

3. Have the authors made all data underlying the findings in their manuscript fully available?

Reviewer #1: No

Reviewer #2: Yes

Reviewer #3: Yes

4. Is the manuscript presented in an intelligible fashion and written in standard English?

Reviewer #1: Yes

Reviewer #2: Yes

Reviewer #3: Yes

5. Review Comments to the Author

Reviewer #1: Thank you for the opportunity to review “Factors associated with domestic violence in the Lahu hill tribe of northern Thailand: A cross-sectional study.” The authors have tackled an important public health issue ie violence against the 3 vulnerable groups. However, I have several pertinent issues of concern with the study/paper, which I will outline below.

i) One of the main concern the study methodology and analysis procedure. The author indicated in their methods section of the paper that the target populations of interest were children (<=15 yrs), women and the elderly. They have also outlined the questionnaire items used to measure violence in each of the 3 groups. The authors then went on to derived a measure for domestic violence from these items. Considering that the drivers of violence in the 3 groups are likely to be different, I would have preferred if the authors conducted a stratified analysis instead so they can look at these groups separately. The type of violence experienced by the 3 groups are different and most likely perpetrated by different members of the HH. Possibly the factors associated with violence experienced by the 3 groups are also different.

ii) Another concern is the study methodology. The authors have not clearly outlined how participants from households were recruited. Did they include everyone in the household (children, mother & father and the elderly)? If they did, did they account for clustering by household in their analysis?

If they did not, then they need to clearly explain in their methodology how HH members were selected.

iii) The authors have defined children as <=15yrs. They need to clearly indicate the minimum age for this group in the methods section (as part of inclusion criteria definition) and how data was collected from them ( eg was it interviewer administered?). I understand in Table 1 they have outlined that the minimum age in the sample was 5yrs. But it is not clear whether 5yrs was a cut off or just minimum age for those interviewed. I am also rather concerned about the integrity of information on violence corrected from children (10yrs or less)

iv) The authors also indicated in the methods section that the target population of interest were children (<=15yrs), women(16-59yrs) and the elderly (>=60yrs). However, their data shows that they also have men (n=382) aged between 16-59yrs in the sample. I do not understand the rationale for including men (16-59yrs) in the study.

v) The general norm for violence studies is to analyze data for male and females separately as they tend to have different risk factors. I would prefer that they present their analysis that way.

Other general comments are:

• They can make their tables more succinct by reporting one category for all binary variables (‘yes”) rather both ‘yes’ and ‘no’

• The manuscript can do with some proof reading and editing.

• The author should consider having prevalence estimates for each type of violence (eg physical violence).

• With the study having 3 groups, it is not clear if participants were reporting their own behavior ( eg alcohol use) and wondering what could be the impact of these on the estimates.

• Sample size formula (if it is necessary to show it) should be explained in the context of the study, e.g what is p in this study?

• They could have a separate table for the multivariate analysis.

Reviewer #2: Background

1.Please clarify the scope of this study because there is different definition between domestic violence and intimate partner violence so the reader might more clear and understand (Domestic violence is violence in the family, the perpetrator is family member, whereas intimate partner violence is violence between partner)

2. The authors should mention research gaps and why have to conduct this study in Lahu tribe.

Method

1. How the researchers randomly selected the village?

2. The sample size calculation is n=826 but the researcher collected the data from n=1,028. Please calrify and give the explanation

3. What is the reliability (pre-test) value from this study?

Result

1. Heading of Table 3 show n= 647 but in detail in the table showed n= 1,028. However, some item "n" is lower than 1,028. Please check the consistency of the data

2. Recoomendation pls consider to give recommendation to 3 level; individual, community, and policy level

Reviewer #3: General comments: Authors have made a good attempt to analyse the extent of domestic violence and factors associated with domestic among children, women and the elderly in a poor community in rural Thailand. This paper will contribute to understanding of domestic violence in low and middle income countries, and provide data for understanding subgroups. However, the authors have introduced interest in studying subgroups but have not done so in the analyses, and this presents a challenge in the analysis and discussion of the generalize results as they often seek to make comparisons or justifications of their results against subgroups elsewhere without actual analyses of their own to back up the comparisons. My recommendations are that the authors must revise their analyses and present the overall study population analysis and present children: girls and boys under 15 years, women and men (by whichever is legal age limit for marriage perhaps), elderly women and men over 60 years. The lack of data on perpetrators is a major limitation but inferences may be safely drawn with supporting study as to who the common perpetrators of DV usually are in such contexts in Thailand. The subgroup analyses will help authors make a case for the understanding of how substance abuse increases risk of violence for specific subgroups to some extent even without understanding the perpetrators data.

Abstract:

Revise according to the recommendations from the content of the manuscript

Background:

“Lead to”: Authors may not reach conclusion that certain variable ‘lead to’ certain risks with cross-sectional data including their own study as such studies only allows for determination of risk factors. We do speak of associations unless experimental studies were done.

“Substance use” versus abuse: authors have described substance ‘use’ and not abuse this makes it difficult to determine risk outright without measuring the extent of misuse.

The background starts well with the global picture but needs to locate the domestic violence problem and risk factors in the context of Thailand, and do so indicating the extent to which other studies have addressed this issue.

- Go into some detail about the Thai social life, income, earnings, religion, culture and describing who Thai people are, what other tribes or social groupings exist in Thailand and the extent of violence there as well as any risk factors or protective factors against DV.

- Also contrast with other neighbouring countries where possible, at least in terms of the prevalence.

- Any cross-border issues that make the hill tribes different from other Thai society?

- Describe the context of substance use and substance abuse in Thailand. How are amphetamines accessed? Natural or other synthetic stimulants? History of substance abuse. What kind of alcohol is used or abused? How is it sourced in communities which have been described as poor? How do they access amphetamines?

- I also propose that some effort into explaining violence against children younger than 15 years be made, and explaining how such studies were done in Thailand or neighbouring countries. What are the risk factors?

Page 3:

Paragraph 1:

- the mention of the SDGs: This is a goal and not a fact, so the authors need to talk about this SDG in the correct context. What are authors trying to say about this goal?

- at the end of paragraph: What do the authors mean ‘domestically’?

Paragraph 2:

- at the end of the paragraph, authors mention contributors to violence: Do authors mean ‘perpetrators’? how do they contribute? If as the perpetrators then authors should state that explicitly. And mention other contributors if they exist. In some context, we have instigators and some Asian studies refer to instigators.

Last paragraph describing hill tribes needs to go into the Setting/study site information in the methods section and explain the following: What is the population size relative to the largest and smallest tribe? Describe the lifestyle that distinguishes Lahu people from others?

Last paragraph of the Background section: Provide a justification for the paper explaining the conceptual framework used for this study as mentioned in the methods section. Thus explaining why a paper on prevalence and associated factors. Why was the study conducted? And why its important for the paper to be published?

Methods:

Paragraph 1 - Explain size of the Lahu hill area. Sources of living such as access to water, income, who is working or not, what is the family structure?

Paragraph 2 - Study sample recruitment needs to be explained. The study aim suggests data was collected for children, women and older persons. It is not clear though what the sampling unit was for the study. Was it families or just individuals in the community?

Paragraph 4 regarding the questionnaire:

- Which questions were inappropriate for the Lahu people due to culture and beliefs?

- Last sentence about pilot is incomplete. Explaining the pilot is good. So finish the sentence please.

Paragraph 5 about the questionnaire:

- Authors need to present a table of each set of questions measured, describing what the variable is made up of, and explaining the source of the variables especially if used in other studies or new including those verified and changed with inputs from the Lahu people.

- Please name that dependent variable and clarify what it comprised of it is a combination of variables. What is the outcome variable made up of?

- Domestic violence questions need to be explained in more detail as an outcome variable. The explanation given here is different compared to what is presented in the results. Clearly explain all variables here first, what they entail including any variation that may be presented in the results. Also provide the time frame that is reported for each form of DV, lifetime experience or in the past 12 months?

Paragraph 6:

- Informed consent forms are meant to remain with the authors particularly to demonstrate that consent was sought and obtained from participants. Informed consent is also a process, it is not clear how this was done. Rather explain how information about the study was provided to participants, what was were these essential aspects that were explained, and how consent was given by particpants.

- Sentence: “People who were not able to provide essential information due to personal health problems were excluded from the study” does not explain the eligibility criteria for sample to participate in the study. Please explain the criteria used, was it one factor or multiple factors that were considered?

- Issue of language used by participants: Move this to the questionnaire section. Was this a paper-based questionnaire? Explain how data was collected in more precisely, like language, who collected the data, how they were trained, their level of qualification. What the role of the village health volunteers including what they usually do.

- How was a confidential room determined? Maybe explain this in a different way? Merge the note about ethical clearance at the end of this paragraph.

Data analysis:

Logistic regression modelling is a process, explain that process. What was the outcome variable, candidate variable, how they were determined first, were bivariate analyses performed first, how were candidate variables selected and fitted into the model, any process of elimination engaged? This is crucial information to determine what was done and whether it can be replicated and reach the same conclusions the authors did.

Results section:

- Table 1 and Table 2 and Table 3 are difficult to decipher because the study is looking at 3 sets of participants: children under 15, women, older persons older than 60.

- Authors need to demonstrate the same characteristics for the study population alongside these characteristics for the study population sets. Including data from men and boys and classifying elders by difference between women, men, girls, boys, older women, older men. The socio-demographics do not make sense without a clear understanding of what the univariate data looks like for all sets of individuals. This is important as data also needs to account for any child marriage which is possible under 15 years, number of family members living in the same household, etc. Vulnerability to violence is also determined by gender. Cross tabulation of the differences by gender is also key in this analysis and table.

- In Table 2 there are other drugs that have been measured which are not explained in the methods section. This justifies why a table including types of substances is needed.

- Table 3: Authors can provide an overall estimation of DV for the population of interest, but still need to extent the analysis to demonstrate the differences between children, women and elderly persons, so that there is clarity on the extent of vulnerability in the population including their subgroups, this is most useful to provide guidance to which groups health interventions should prioritise and address and what actual interventions could be considered for which subgroups or to address which factors associated with vulnerability of which subgroup. Lumping all children, women and elderly persons is not justified as an end in itself, but would be more valuable if authors are able to single out the most vulnerable, even at a comparative level. The analysis is insufficient at this level and needs to demonstrate the complexities that are entailed in domestic violence as a multi-layered construct that can be differentiated for women into intimate partner violence or domestic violence by family members other than husband if women are married, or child abuse by parents or other family members, or even strangers if the latter was measured, etc.

- In Table 3, for each type of domestic violence need to provide a composite estimate as done with domestic violence. This will explain the extent of physical or emotional abuse overall.

- Authors need to explain the risk of smoking and how it links to perpetration of violence, as this variable doesn’t make sense in terms of how it increases vulnerability to DV. Do this in the background section.

- Table 4: The analysis appears to have skipped a stage as not all variables need to go into the multivariate analysis, so authors need to demonstrate candidate variables through bivariate analysis and look at some degree of association of each variable with the outcome variable. So bivariate analysis is required. Authors much explain Why are some analyses adjusted for and others are not?

Discussion:

- The authors are evidently struggling to explain the study prevalence in it entirety without a clear indication per children, women, elderly women or men subsets analysed. The data does not provide data to indicate the extent of violence against children. The story of substance use in general is also not telling a clear story without understanding the extent to which increased risk and perhaps authors should also report on who is using, and whether combinations of substances are used and determine risk from that angle. However, the conclusions reached in this paper cannot be reached with the currently presented analyses. Authors need to address the analytical questions about extent of vulnerability to violence per subgroup, what increases their vulnerability in most precise terms than the general picture painted already which does not in itself make much sense. Then tell the role substances play in this story.

- Paragraph 2: These comparisons are incorrect without a clear understanding of the prevalence of DV among women from your analysis, so this is incorrect comparison. The same point applies to comparisons on data about children or the elderly. The analyses failed to demonstrate prevalence for each subgroup so there cannot be comparisons with subgroups.

- Paragraph 3: This is suggesting that Buddist society is more gender equitable, where is the data to support this? The argument does not make sense as patriarchy is a common factor across all societies. Authors need to provide a clear explanation of the religious differences from the background and link their interpretation to that. It may be that there is a vast difference based on religions but the current argument to explain these results is incorrect without studies to support it. Authors need to explain what they mean by ‘complicated relationships’. How is Buddhism organized and are relationships there complicated too or not?

Conclusions:

The conclusions need to be strengthened and be drawing out what the results have demonstrated about the study sample as sub-groups and as a whole, and indicating more specifically what recommendations suite which aspect of the results.

Abbreviations:

Authors must integrate in the texts and ensure each is explained in full at first mention and then abbreviated afterwards.

Ethical considerations

- must be integrated with the details on how consent was obtained and data collected. Add a little bit on what DV specific considerations were made? What guidelines were followed for conducting these interviews on such a sensitive subject? Explain data storage.

References:

Many of the references do not follow the journal’s preferred referencing style. Visit: https://journals.plos.org/plospathogens/s/submission-guidelines#loc-references for information on Vancouver style. Here is some simpler information about what you can do and examples provided.

PLOS uses the reference style outlined by the International Committee of Medical Journal Editors (ICMJE), also referred to as the “Vancouver” style. Example formats are listed below. Additional examples are in the ICMJE sample references. A reference management tool, EndNote, offers a current style file that can assist you with the formatting of your references. If you have problems with any reference management program, please contact the source company's technical support.

6. PLOS authors have the option to publish the peer review history of their article (what does this mean?). If published, this will include your full peer review and any attached files.

Reviewer #1: No

Reviewer #2: No

Reviewer #3: No

---

## [Author Response · Author response to Decision Letter 0]

27 Dec 2020

Response go reviewers’ comments

Dear Editor,

We have used long time in re-anaysis and re-forming the presentation, checking in whole text and response to all comments of reviewers. Some comments, we could not response because there is no information available. However, we very hope that you and reviewers would be happy in this version.

Finally, we sorry that using a longer time to be able to response to your comments.

Best regards,

TK

Editor comment

We have received comments from three reviewers. One reviewer recommended rejection, one recommended minor revisions, and the other one recommended major revisions. The primary concerns are around the sampling and data analysis methods. I agree with two reviewers that stratified analyses should have been performed to explore the differences in domestic violence prevalence and its risk factors in different sub-populations. The sub-populations included girls and boys under 15 years, women and men, and female and male elderly over 60 years. The authors expressed an interest in studying the subgroups and should have reflected this in the analyses. The lack of sub-group results introduced a challenge in discussing and comparing this study's findings against other studies. We will consider this manuscript for further review if the authors could address these concerns.

1.We suggest you thoroughly copyedit your manuscript for language usage, spelling, and grammar. If you do not know anyone who can help you do this, you may wish to consider employing a professional scientific editing service. 

: Thank you very much for the points raised, we have a lot concern about that. The manuscript is checked and re-edited by the American Journal Experts. 

2. Please include a copy of the questionnaire, in the original language, as Supporting Information.

: Thank you, it’s attached (Thai version).

3. In statistical methods, please clarify whether you corrected for multiple comparisons.

: We revised the whole section of data analysis. There are divided into three groups; children (5-15 years), women (aged 16-59 years), and elderly (aged 60 years and over).

All steps of the analysis, we sued the multiple logistic regression to compare between who had violence and did not to fine the final model before interpreting the factors associated with domestic violence in each group.

4. Cross-sectional study design cannot be used to determine causation. In this light, please revise your conclusions to indicate that your study presents associations between exposure variables and domestic violence. For example, your statement that "smoking and alcohol and amphetamine use, leads to domestic violence" implies causation which is cannot be determined using your study design.

: Thank you very much for such great comment. All points have been revised including the mathematic forms used in abstract, results, and discussion (page 2; lines 14-24; page 11, lines 7-15. 

5. Please amend your list of authors on the manuscript to ensure that each author is linked to an affiliation. Authors’ affiliations should reflect the institution where the work was done (if authors moved subsequently, you can also list the new affiliation stating “current affiliation:….” as necessary).

: Thank you, we have done in double checked.

: Thank you so much, it has been moved into the proper place (page 6, line 10-16). All information in the ethical statement is the same put into the system during submission the manuscript.

: Thank you very much for the suggestion. All relevant supporting information have been placed in a proper place in the manuscript with corresponding to the journal guideline.

Reviewer #1: Thank you for the opportunity to review “Factors associated with domestic violence in the Lahu hill tribe of northern Thailand: A cross-sectional study.” The authors have tackled an important public health issue ie violence against the 3 vulnerable groups. However, I have several pertinent issues of concern with the study/paper, which I will outline below.

i) One of the main concern the study methodology and analysis procedure. The author indicated in their methods section of the paper that the target populations of interest were children (<=15 yrs), women and the elderly. They have also outlined the questionnaire items used to measure violence in each of the 3 groups. The authors then went on to derived a measure for domestic violence from these items. Considering that the drivers of violence in the 3 groups are likely to be different, I would have preferred if the authors conducted a stratified analysis instead so they can look at these groups separately. The type of violence experienced by the 3 groups are different and most likely perpetrated by different members of the HH. Possibly the factors associated with violence experienced by the 3 groups are also different.

: Thank you so much for this valuable comment. We totally agree with you and then making different procedure for the analysis (three different group, three stratifies). 

: The results show very interestingly in table 4 (violence to women), table 5 (violence to children), and table 6 (violence to the elderly) 

ii) Another concern is the study methodology. The authors have not clearly outlined how participants from households were recruited. Did they include everyone in the household (children, mother & father and the elderly)? If they did, did they account for clustering by household in their analysis? If they did not, then they need to clearly explain in their methodology how HH members were selected.

: All eligible populations (Children aged 5-15 years, women aged 16-59 years, and elderly 60 years and above) who live in 20 selected Lahu villages were invited to the study. 

iii) The authors have defined children as <=15yrs. They need to clearly indicate the minimum age for this group in the methods section (as part of inclusion criteria definition) and how data was collected from them (eg was it interviewer administered?). I understand in Table 1 they have outlined that the minimum age in the sample was 5yrs. But it is not clear whether 5yrs was a cut off or just minimum age for those interviewed. I am also rather concerned about the integrity of information on violence corrected from children (10yrs or less)

: We intended to collect data from children 5-15 years, which come from our 13 years experience in doing research among the hill tribe. Below 5 years is very difficult to obtain the validate information. 

iv) The authors also indicated in the methods section that the target population of interest were children (<=15yrs), women(16-59yrs) and the elderly (>=60yrs). However, their data shows that they also have men (n=382) aged between 16-59yrs in the sample. I do not understand the rationale for including men (16-59yrs) in the study.

: We so sorry for the mistake of process of case selection during the analysis. We have mad revised in whole process of data analysis. 

v) The general norm for violence studies is to analyze data for male and females separately as they tend to have different risk factors. I would prefer that they present their analysis that way.

: Thank you very much for the comment. We have now making three subcategories for the analysis; children aged 5-15 years, women 16-59 years, and the elderly aged 60 years and over. The whole characteristics and other information have been separated. 

Other general comments are:

• They can make their tables more succinct by reporting one category for all binary variables (‘yes”) rather both ‘yes’ and ‘no’

: Thank you for the comment, we have tried to make the best option, and present in Yes and No to confirm the readers on the validation of the presentation. We very hope you understand us.

• The manuscript can do with some proof reading and editing.

: Thank you so much, it was re-edited by the American Journal Experts.

• The author should consider having prevalence estimates for each type of violence (eg physical violence).

: Yes, the data are presented in the results section.

• With the study having 3 groups, it is not clear if participants were reporting their own behavior ( eg alcohol use) and wondering what could be the impact of these on the estimates.

: We have classified all items or factors according to three groups, in table 2. Very hope it helps.

• Sample size formula (if it is necessary to show it) should be explained in the context of the study, e.g what is p in this study?

: Thank you, we have carefully explain detail, please see page 4-5.

• They could have a separate table for the multivariate analysis.

: Yes, we did. There are now separated into different table, table 4 (children), 5 (women), and 6(elderly)

Reviewer #2: Background

1.Please clarify the scope of this study because there is different definition between domestic violence and intimate partner violence so the reader might more clear and understand (Domestic violence is violence in the family, the perpetrator is family member, whereas intimate partner violence is violence between partner)

: Yes, we intended to understand the domestic violence which is defined as a pattern of behavior in any relationship that is used go gain or maintain power and control over a family members. This definition is followed the UN definition of domestic violence. Page 6.

2. The authors should mention research gaps and why have to conduct this study in Lahu tribe.

: We did, page 4, lines 7-10.

Method

1. How the researchers randomly selected the village?

: We did selection the village by which were selected by a computer generate random number method, page 4, lines 17-18.

2. The sample size calculation is n=826 but the researcher collected the data from n=1,028. Please calrify and give the explanation

: We so sorry, some procedure in sample size calculation was incorrect. We have made correction, page 4 line 27-31, and page 5, line 1-5.

3. What is the reliability (pre-test) value from this study?

 : The overall Cronbach alpha was 0.71, page 6, line 4.

Result

1. Heading of Table 3 show n= 647 but in detail in the table showed n= 1,028. However, some item "n" is lower than 1,028. Please check the consistency of the data

: So sorry for the mistake, all data in all table have been double checked and revised.

2. Recoomendation pls consider to give recommendation to 3 level; individual, community, and policy level

: We have added in conclusion section, page 28, lines 12-25.

Reviewer #3: General comments: Authors have made a good attempt to analyse the extent of domestic violence and factors associated with domestic among children, women and the elderly in a poor community in rural Thailand. This paper will contribute to understanding of domestic violence in low and middle income countries, and provide data for understanding subgroups. However, the authors have introduced interest in studying subgroups but have not done so in the analyses, and this presents a challenge in the analysis and discussion of the generalize results as they often seek to make comparisons or justifications of their results against subgroups elsewhere without actual analyses of their own to back up the comparisons. My recommendations are that the authors must revise their analyses and present the overall study population analysis and present children: girls and boys under 15 years, women and men (by whichever is legal age limit for marriage perhaps), elderly women and men over 60 years. The lack of data on perpetrators is a major limitation but inferences may be safely drawn with supporting study as to who the common perpetrators of DV usually are in such contexts in Thailand. The subgroup analyses will help authors make a case for the understanding of how substance abuse increases risk of violence for specific subgroups to some extent even without understanding the perpetrators data.

: Thank you so much, we have completely done on classifying to analysis the factors associated with three different groups; children (table 4), women (table 5), and elderly (table 6)

Abstract:

Revise according to the recommendations from the content of the manuscript

: Thank you, it’s revised and improved. 

Background:

“Lead to”: Authors may not reach conclusion that certain variable ‘lead to’ certain risks with cross-sectional data including their own study as such studies only allows for determination of risk factors. We do speak of associations unless experimental studies were done.

: Thank you, we have changed as comment. 

“Substance use” versus abuse: authors have described substance ‘use’ and not abuse this makes it difficult to determine risk outright without measuring the extent of misuse.

: Thank you, it’s changed properly in whole text.

The background starts well with the global picture but needs to locate the domestic violence problem and risk factors in the context of Thailand, and do so indicating the extent to which other studies have addressed this issue.

: Thank you,

: We have revised and mentioned many risk factors related to domestic violence in Thailand and its references, page 3, line 20-29

- Go into some detail about the Thai social life, income, earnings, religion, culture and describing who Thai people are, what other tribes or social groupings exist in Thailand and the extent of violence there as well as any risk factors or protective factors against DV.

: We have greatly investigated on the information between Thailand and the hill tribe, unfortunately there is little information found, and no information presented in protective factor. We would say that no-study had been done previously about domestic violence in the hill tribe. So sorry.

- Also contrast with other neighbouring countries where possible, at least in terms of the prevalence.

: We have explored the information, unfortunately there is no scientific information regard to domestic violence among tribe people in Myanmar or Laos as well.

: Information is presented to non-scientific writer which is mostly not relevant to domestic violence.

- Any cross-border issues that make the hill tribes different from other Thai society?

: The different of the hill tribe and Thai people is not from cross-border but rather from the difference of their culture and other life styles.

- Describe the context of substance use and substance abuse in Thailand. How are amphetamines accessed? Natural or other synthetic stimulants? History of substance abuse. What kind of alcohol is used or abused? How is it sourced in communities which have been described as poor? How do they access amphetamines?

: We have added in reference 16-17. Even Thailand has been reported on having amphetamine use, but surprising there is very few scientific papers available. 

- I also propose that some effort into explaining violence against children younger than 15 years be made, and explaining how such studies were done in Thailand or neighbouring countries. What are the risk factors?

: We have added some information in the discussion part. However, there is very few paper published. 

Page 3:

Paragraph 1:

- the mention of the SDGs: This is a goal and not a fact, so the authors need to talk about this SDG in the correct context. What are authors trying to say about this goal?

- at the end of paragraph: What do the authors mean ‘domestically’?

: Thank you very much for the great comment.

Paragraph 2:

- at the end of the paragraph, authors mention contributors to violence: Do authors mean ‘perpetrators’? how do they contribute? If as the perpetrators then authors should state that explicitly. And mention other contributors if they exist. In some context, we have instigators and some Asian studies refer to instigators.

: We are trying to say that under the context of Thai society which men dominate in family, including the referred paper said that the men was defined as the domestic makers.

: We sorry if the response is not going to what your questions. If you could make more easier would be great appreciated.

Last paragraph describing hill tribes needs to go into the Setting/study site information in the methods section and explain the following: What is the population size relative to the largest and smallest tribe? Describe the lifestyle that distinguishes Lahu people from others?

: Thank you for the comment. We studied in the Lahu because one of our small previous study presented that their have a large proportion who used alcohol. Moreover, some information were found on the domestic violence among the Lahu people. This become our current study. Unfortunately, we do not have information about very some sensitive behaviors such as substances use including domestic violence in other tribes. Then, we decide not to mention the information of other tribes. We so sorry, we will do more study before could be able to come up the scientific information these points.

Last paragraph of the Background section: Provide a justification for the paper explaining the conceptual framework used for this study as mentioned in the methods section. Thus explaining why a paper on prevalence and associated factors. Why was the study conducted? And why its important for the paper to be published?

: Thank you so much for such great comments here.

: We have modified, improved, and revised as your comments. Page 4, lines 7-10.

Methods:

Paragraph 1 - Explain size of the Lahu hill area. Sources of living such as access to water, income, who is working or not, what is the family structure?

: Thank you, it’s revised in whole paragraph. Page 4-5.

Paragraph 2 - Study sample recruitment needs to be explained. The study aim suggests data was collected for children, women and older persons. It is not clear though what the sampling unit was for the study. Was it families or just individuals in the community?

: Thank you, it’s revised and explained in page 6, line 14-28.

Paragraph 4 regarding the questionnaire:

- Which questions were inappropriate for the Lahu people due to culture and beliefs?

: Do you kick your wife? In this question, we can ask directly. Many questions, they did not answer if they did not trust interviewer.

: Do you think you have a problem with your husband in amphetamine use?

- Last sentence about pilot is incomplete. Explaining the pilot is good. So finish the sentence please.

: Thank you and sorry for the mistake. It’s improved.

Paragraph 5 about the questionnaire:

- Authors need to present a table of each set of questions measured, describing what the variable is made up of, and explaining the source of the variables especially if used in other studies or new including those verified and changed with inputs from the Lahu people.

: We used our own questionnaire developed. The detail is explained in page 5 and 6

: The detail of questionnaire both Thai and English have been attached.

- Please name that dependent variable and clarify what it comprised of it is a combination of variables. What is the outcome variable made up of?

: DV is domestic violence which is very special forms occurred in Lahu family, see page 6, lines 4-12.

: We have known the forms from our pilot step. All forms have been developed to questions which present the result in Table 3. 

- Domestic violence questions need to be explained in more detail as an outcome variable. The explanation given here is different compared to what is presented in the results. Clearly explain all variables here first, what they entail including any variation that may be presented in the results. Also provide the time frame that is reported for each form of DV, lifetime experience or in the past 12 months?

: It’s improved and explained in page 6, line 5-12.

Paragraph 6:

- Informed consent forms are meant to remain with the authors particularly to demonstrate that consent was sought and obtained from participants. Informed consent is also a process, it is not clear how this was done. Rather explain how information about the study was provided to participants, what was were these essential aspects that were explained, and how consent was given by particpants.

: Thank you, it’s explained in page 6, lines 11-16. 

- Sentence: “People who were not able to provide essential information due to personal health problems were excluded from the study” does not explain the eligibility criteria for sample to participate in the study. Please explain the criteria used, was it one factor or multiple factors that were considered?

: Thank you, it’s explained. Page 4, lines 24-25.

- Issue of language used by participants: Move this to the questionnaire section. Was this a paper-based questionnaire? Explain how data was collected in more precisely, like language, who collected the data, how they were trained, their level of qualification. What the role of the village health volunteers including what they usually do.

: Thank you, we were helped from the village health volunteers in filling general information. However, those who could not read Thai, the forms were filled by psychiatric nurse with the help of community heath volunteers,

- How was a confidential room determined? Maybe explain this in a different way? Merge the note about ethical clearance at the end of this paragraph.

Data analysis:

: A confident room is provided by village headmen at the community hall. Page 6, line 22-23.

Logistic regression modelling is a process, explain that process. What was the outcome variable, candidate variable, how they were determined first, were bivariate analyses performed first, how were candidate variables selected and fitted into the model, any process of elimination engaged? This is crucial information to determine what was done and whether it can be replicated and reach the same conclusions the authors did.

: The DV is the having domestic violence (Y/N). Then, it forces us to us the logistic regression for the analyses. The stepe of the analysis, we used “Enter” mode which is allowed researcher to consider a IV put into the model because we have to concern on statistic significant and public health significant. 

: Detail have been explained in page 6, line 30-31, and page 7 lines, 1-4.

Results section:

- Table 1 and Table 2 and Table 3 are difficult to decipher because the study is looking at 3 sets of participants: children under 15, women, older persons older than 60.

: Thank you, we have revised in analyses and interpretations in whole set of data. Please see table 1-6 which are classified into three main groups.

- Authors need to demonstrate the same characteristics for the study population alongside these characteristics for the study population sets. Including data from men and boys and classifying elders by difference between women, men, girls, boys, older women, older men. The socio-demographics do not make sense without a clear understanding of what the univariate data looks like for all sets of individuals. This is important as data also needs to account for any child marriage which is possible under 15 years, number of family members living in the same household, etc. Vulnerability to violence is also determined by gender. Cross tabulation of the differences by gender is also key in this analysis and table.

: We agree with you, and did reanalysis in the whole set of data, please see table 1, page 7.

- In Table 2 there are other drugs that have been measured which are not explained in the methods section. This justifies why a table including types of substances is needed.

: From our pilot stage, most of substances use among the Lahu had been clarify, then we intended to measure only this 6 items. 

- Table 3: Authors can provide an overall estimation of DV for the population of interest, but still need to extent the analysis to demonstrate the differences between children, women and elderly persons, so that there is clarity on the extent of vulnerability in the population including their subgroups, this is most useful to provide guidance to which groups health interventions should prioritise and address and what actual interventions could be considered for which subgroups or to address which factors associated with vulnerability of which subgroup. Lumping all children, women and elderly persons is not justified as an end in itself, but would be more valuable if authors are able to single out the most vulnerable, even at a comparative level. The analysis is insufficient at this level and needs to demonstrate the complexities that are entailed in domestic violence as a multi-layered construct that can be differentiated for women into intimate partner violence or domestic violence by family members other than husband if women are married, or child abuse by parents or other family members, or even strangers if the latter was measured, etc.

: Defiantly, we have revised the analysis in table 3, please see pages 11-12.

- In Table 3, for each type of domestic violence need to provide a composite estimate as done with domestic violence. This will explain the extent of physical or emotional abuse overall.

: Thank you, it’s revised and improved.

- Authors need to explain the risk of smoking and how it links to perpetration of violence, as this variable doesn’t make sense in terms of how it increases vulnerability to DV. Do this in the background section.

: We have modified the discussion in this part and added some information. Page 26, lines 1-12.

- Table 4: The analysis appears to have skipped a stage as not all variables need to go into the multivariate analysis, so authors need to demonstrate candidate variables through bivariate analysis and look at some degree of association of each variable with the outcome variable. So bivariate analysis is required. Authors much explain Why are some analyses adjusted for and others are not?

: Sorry, please forget the old style of the analysis and presentations. We did re-analysis in classifying into three groups, please kindly see table 4-6 in new results. 

Discussion:

- The authors are evidently struggling to explain the study prevalence in it entirety without a clear indication per children, women, elderly women or men subsets analysed. The data does not provide data to indicate the extent of violence against children. The story of substance use in general is also not telling a clear story without understanding the extent to which increased risk and perhaps authors should also report on who is using, and whether combinations of substances are used and determine risk from that angle. However, the conclusions reached in this paper cannot be reached with the currently presented analyses. Authors need to address the analytical questions about extent of vulnerability to violence per subgroup, what increases their vulnerability in most precise terms than the general picture painted already which does not in itself make much sense. Then tell the role substances play in this story.

: Thank you, you are right. We have made revision in whole paragraph after getting information from the analysis in three groups. 

- Paragraph 2: These comparisons are incorrect without a clear understanding of the prevalence of DV among women from your analysis, so this is incorrect comparison. The same point applies to comparisons on data about children or the elderly. The analyses failed to demonstrate prevalence for each subgroup so there cannot be comparisons with subgroups.

: Yes, we agree with you. We have revised in this whole paragraph after getting the three groups analyses.

- Paragraph 3: This is suggesting that Buddist society is more gender equitable, where is the data to support this? The argument does not make sense as patriarchy is a common factor across all societies. Authors need to provide a clear explanation of the religious differences from the background and link their interpretation to that. It may be that there is a vast difference based on religions but the current argument to explain these results is incorrect without studies to support it. Authors need to explain what they mean by ‘complicated relationships’. How is Buddhism organized and are relationships there complicated too or not?

: Unfortunately, after carefully re-analysis in three different groups, religion was not found the association with domestic violent. Then, it’s deleted from the sdicussion.

Conclusions:

The conclusions need to be strengthened and be drawing out what the results have demonstrated about the study sample as sub-groups and as a whole, and indicating more specifically what recommendations suite which aspect of the results.

: Thank you, It’s revised and improved.

Abbreviations:

Authors must integrate in the texts and ensure each is explained in full at first mention and then abbreviated afterwards.

: Thank you, we have checked all relevant text.

Ethical considerations

- must be integrated with the details on how consent was obtained and data collected. Add a little bit on what DV specific considerations were made? What guidelines were followed for conducting these interviews on such a sensitive subject? Explain data storage.

: Thank you for the comment, page 6, lines 4-12.

: the ethical has been moved to methods section. The content is also extended. 

References:

Many of the references do not follow the journal’s preferred referencing style. Visit: https://journals.plos.org/plospathogens/s/submission-guidelines#loc-references for information on Vancouver style. Here is some simpler information about what you can do and examples provided.

PLOS uses the reference style outlined by the International Committee of Medical Journal Editors (ICMJE), also referred to as the “Vancouver” style. Example formats are listed below. Additional examples are in the ICMJE sample references. A reference management tool, EndNote, offers a current style file that can assist you with the formatting of your references. If you have problems with any reference management program, please contact the source company's technical support.

: Thank you so much, all references have been re-checked and re-formatting.

---

## [Decision Letter · Decision Letter 1]

22 Jan 2021

PONE-D-20-27497R1

Factors associated with domestic violence in the Lahu hill tribe of northern Thailand: A cross-sectional study

PLOS ONE

Dear Dr. Apidechkul,

Thank you for submitting your manuscript to PLOS ONE. After careful consideration, we feel that it has merit but does not fully meet PLOS ONE’s publication criteria as it currently stands. Therefore, we invite you to submit a revised version of the manuscript that addresses the points raised during the review process.

We look forward to receiving your revised manuscript.

Kind regards,

Siyan Yi, MD, MHSc, PhD

Academic Editor

PLOS ONE

Additional Editor Comments:

**Editor’s comments**

We thank the authors for addressing outstanding comments from the reviewers. The revised manuscript has been improved. However, the English writing quality remains well below an acceptable level, particularly. I have read the entire manuscript; however, my comments are not exhaustive. The manuscript requires substantial support from an experienced English writer with sufficient understanding of the research context.

In your response to reviewers, please provide the details on how you addressed the comments (what changes were made, where in the text are revised), not just saying those particular points were revised and improved (but what were revised or improved and where). Where data were reanalyzed, explain briefly the new findings and ensure that revisions have been made to reflect the new findings (what were added or removed and where). To most of the comments, particularly those from Reviewer #3, the responses are not sufficient without clear explanations. In your revised manuscript preparation, please have a closer look at the Instructions for Authors and follow them strictly, including word count limit, styles, spacing adjustments, etc.

**Abstract**

The abstract must be condensed as it contains far more than the recommended 350 word count limit.Line 26, page 1: Please remove ‘sexual problems’ as it is outside of the study’s scope and is not applicable to all target populations under the study (e.g., children).Please use consistent terminologies; e.g., ‘the elderly’ in place of ‘elderly,’ ‘elderly adults,’ ‘elderly individuals,’ ‘older people,’ ‘elderly persons,’ etc. throughout the text.Line 26, page 1: ‘types of’ is unnecessary.Line 30, page 1: The word ‘adults’ is unnecessary.Line 1, page 2: A cross-sectional study was conducted among participants who belonged to the Lahu hill tribe and lived in 20 selected villages in Chiang Rai Province, Thailand.Lines 2-5: A validated questionnaire was used to collect personal information and experiences related to domestic violence in the past year from children aged 5-15 years, women (aged 16-59 years), and elderly aged ³60 years.Line 5, page 2: Please specify whether the ‘logistic regression analyses’ were bivariate, multiple, or both.Line 6, page 2: Please remove ‘at the significance level µ=0.05.’Results section must be shortened, removing most of the descriptive socio-demographic results.The word ‘analysis’ is a countable noun, and its plural form should be ‘analyses.’Line 10, page 2: The use of ‘while’ is incorrect.Line 12, page 2: …children who smoked…Line 15, page 2: …a family head…Lines 14-20, page 2: The large sentence must be restructured as it is difficult to read. It may be broken down into small sentences. Also, when you present the associations between variables, please make the significance level clear, not just say ‘more likely.’

**Introduction**

Line 7, page 3: In scientific writing, numbers smaller than 10 should be spelt out.Line 19, page 3: …the elderly living in developing countries…Lines 21-23, page 3: Please revise the sentence to minimize wording redundancies (children, women, the elderly) and improve the referencing. The references 11, 12, and 13 should be moved to join 14 and 15 at the end of the sentence.Line 24, page 3: Please remove ‘who were’ as it is unnecessary.Lines 26-27, page 3: “Given people’s economic constraints and substance use, alcohol use has become an integrated factor contributing to domestic violence [18].” This sentence is difficult to understand – how is the first part linked to the latter?Line 8, page 4: Add ‘,’ before ‘including.’

**Methods**

Lines 16-17, page 4: The first and second paragraph should be combined as the first one does not make much sense. Consider: “A cross-sectional study was conducted in 20 villages randomly selected out of 216 Lahu villages in Chiang Rai Province, Thailand [21], using a random number generation method.”Lines 18-19, page 4: Please remove “However, in the study process, only 20 Lahu villages were randomly selected from the list” as does not add any meaning to the text and sounds redundant.Lines 19-20, page 4: The sentence, “All the Lahu people who were living in the selected 20 villages were invited to participate in the study” is not true as you included only children aged 5-15, women, and the elderly.Line 22, page 4: Again, the same sentence was repeated, which should also be removed.Lines 24-25, page 4: The sentence “People who were able to provide essential information regarding the study protocol, living selected villages at the date of data collection were eligible for the study” is not readable. Why should the target populations be able to provide essential information regarding the study protocol? Living in the selected villages? The inclusion and exclusion criteria must be clearly stated.Line 27, page 4: The study sample size…Line 1, page 5: “Therefore, the minimum required sample size for the study was 196 participants for each sub-population (children aged 5-15, women aged 16-59, and the elderly aged ³60).” Then you can remove the whole following sentence (However, in this study, three vulnerable groups (children aged 15 years and below, women aged 16-59, and elderly aged 60 year and over) were selected for the study, then, a total of 588 participants were required.”Line 7, page 5: Did this study have a conceptual framework? If so, it must be described in the text. If not, this expression about it should be removed.Lines 13-15, page 5: The complex sentence needs to be re-structured by breaking it down into a few sentences or using correct punctuations to link the sub-sentences.Lines 17-21, page 5: The long sentence should be rewrite as it is difficult to read and with grammatical errors.Lines 25-26, page 5: Why was substance use limited to only cigarette, alcohol, and amphetamine use? Also, how is cigarette smoking related to domestic violence?Lines 27-28, page 5: Are sexual harassment and being forced to have sex considered harm related to sex, not sexual violence?Line 30, page 5: The word ‘ignorant’ may have been incorrectly used.Lines 1, page 6: Is ‘domestic abuse’ different from ‘domestic violence?’ It is the first mention of domestic abuse in the paper. Also, please say ‘women aged 16-59’ consistently.Line 4, page 6: ‘Cronbach’s alpha.’ Also, state clearly on what is the Cronbach’s alpha for? Domestic violence scale? Across the three participant groups? References to support the scales used in this study must be cited. The use of the scales (coding, recoding, cut-offs used to define the variables…), should be clearly described.Several grammatical mistakes are easily found in the questionnaire development and data collection section. Please revise them carefully. For example, the following three sentences are all incorrect: did you have experienced of kicking out of the house from family member in the past year? Did you have experienced of being forced to drink alcohol, smoke, or use substances in the past year from family member? Did you have experienced of being forced to ask for money or borrow items from others?Please make ‘statistical/data analysis’ plural (data analyses).Line 1, page 7: Please specify whether the ‘logistic regression analyses’ were bivariate, multiple, or both. The description of the variable selection for the multiple regression analysis models is hard to understand. The sentences “Some variables were controlled the effect in the model which were determined as the confounder factors for the prediction. In the final model, all significant variables and controlled variables were fitted before making interpretations” are unreadable.The paragraph on the ethics also needs improvement in writing quality. Did this study need more than one protocol? If so, why? Please correct the wording – e.g., what you need from the participants was their consent, not the form. Please also correct several grammatical errors in the paragraph.

**Results**

All tables and narratives of the tables need to be revised.

Lines 19-20, page 7: Please clarify if the 646 was the number of participants recruited or included in the analyses. If the number of the participants recruited was different from that in the analyses, please provide the details of the number of participants excluded with reasons.Line 22-23, page 7: Please remove ‘While,’ which was incorrectly used, and change ‘elderly persons’ to ‘the elderly.’ Also, please use ‘;’ to separate full sentences, not ‘,’.Lines 21-24, page 7: You don’t need to repeat the number of participants in each group as it was clearly stated in the first sentence.Table 1:Please do not use ‘.’ in titles.The use of ‘;’ is also incorrect.The table’s head is not understandable – Why was the head repeated? Please have a look at other published papers and revise them accordinglyLine 2-5, page 9: Please correct all the sentences in this paragraph, which are unreadable and grammatically incorrect.Table 2:The title is incomplete.The contents of the table are not matched with the table’s head – while the head notes n (%) for Yes and No, one column presents number, the other presents %. Also, it’s unnecessary to present both ‘Yes’ and ‘No’ results. Let readers do the math.While only smoking, alcohol use, and amphetamine use was mentioned in the methods, other types of substances are shown in the table. Please make them consistent. Please also include the last item, ‘Family member who used alcohol’ in the measurements in the methods.The narrative of Table 3 also needs improvement, correcting the misuse of punctuations typos, and inconsistencies. Also, only one category should be presented in tables for binary variables.Line 1, page 14: Please use ‘bivariate logistic regression’ in place of ‘univariate analysis.’ Univariate analyses involve only one variable without a comparison or association and are used to describe participants’ characteristics (e.g., mean, median, %, etc.). Also, ‘smoking behavior’ is not a correct term in this context – you just asked whether they smoked (yes or no), not exploring the way they smoked.In description of relationships, please specify the direction and significance level of the associations, not just saying the variables are associated.Since the description of the regression models was not at all clear, the reader cannot understand why only religion and education were controlled for in Table 4. Same for Tables 5 and 6. Also, please use correct terminologies (e.g., multiple logistic regression analysis).Lines 2-5, page 14: The two sentences tell the same thing and should be combined. This is also applied to Table 5 and 6.Data in Table 4 tell clearly that the sample sizes were not calculated to represent each study population. As a result, the it was too small for children.As commented earlier, please keep only one head for all tables.The footnotes under Tables 4-6 are not understandable – why were µ values needed to tell a significance level, while p-values were already presented in the tables?

**Discussion**

In general, the discussion section requires extensive support from an experience writer. Many statements are not understandable because of the poor quality, many others are too broad and not specifically based on the study findings. The discussions are mostly the comparisons with other studies (A supports B, A is confirmed by B), without context-specific interpretation and linkage to social policy implications.Lines 12-14, page 26: The sentence “The WHO [28] reported that 30.0% of domestic violence against women had different prevalence estimates in different communities that were higher than the estimates in the Lahu community in Thailand” is not at all clear.Line 16, page 26: Please clarify what ‘clinical populations’ means?Lines 1-12, page 27: The discussion on the relationship between smoking and domestic violence relied so much on personal assumptions that are not well supported by the findings or literature.Lines 13-18, page 27: The paragraph almost unreadable at all due to a poor structure and grammatical mistakes.Lines 1-2, page 28: What does ‘…domestic violence was dominated by alcohol use…’ mean? Alcohol use may be a cause of domestic violence, but how it dominates the violence is not understandable.Limitations are not well presented.The sentence does not make sense – the study’s limitations are not determined by the refusal rate.It’s not clear how and to what extend data collection by community health volunteers affected the data quality? What measures were taken to address the concerns? Did the volunteers receive any training? Information on data collection training should be clearly provided in the Methods.Same for the following sentence. Please discuss clearly what were the methodological issues around the collection of sensitive data?Conclusion also need to be entirely re-written. Please avoid repeating the results or include further discussion. Instead, please summarize key findings and provide relevant recommendations that are well supported by the findings.

**References**

The reference list does not meet the journal’s requirements. Please rework on it more carefully.

Reviewers' comments:

Reviewer's Responses to Questions

**Comments to the Author**

1. If the authors have adequately addressed your comments raised in a previous round of review and you feel that this manuscript is now acceptable for publication, you may indicate that here to bypass the “Comments to the Author” section, enter your conflict of interest statement in the “Confidential to Editor” section, and submit your "Accept" recommendation.

Reviewer #2: All comments have been addressed

2. Is the manuscript technically sound, and do the data support the conclusions?

Reviewer #2: Yes

3. Has the statistical analysis been performed appropriately and rigorously? 

Reviewer #2: Yes

4. Have the authors made all data underlying the findings in their manuscript fully available?

Reviewer #2: Yes

5. Is the manuscript presented in an intelligible fashion and written in standard English?

Reviewer #2: Yes

6. Review Comments to the Author

Reviewer #2: Accept. The authors have adequately addressed the comments raised in a previous round of review. This manuscript is now acceptable for publication.

7. PLOS authors have the option to publish the peer review history of their article (what does this mean?). If published, this will include your full peer review and any attached files.

Reviewer #2: No

---

## [Author Response · Author response to Decision Letter 1]

19 Feb 2021

Response to Editor’s comments

Dear Editor,

I do not know how to thank you for the help along the paper. I have learnt a lot from your comments in each item. Your advices comments make us improvement all skills and practices in how to write a good academic paper.

After improvement, the paper has been once again checked by the American Journal Experts. 

I do very hope that you happy in this version.

Thank you once again,

TK

Additional Editor Comments:

Editor’s comments

We thank the authors for addressing outstanding comments from the reviewers. The revised manuscript has been improved. However, the English writing quality remains well below an acceptable level, particularly. I have read the entire manuscript; however, my comments are not exhaustive. The manuscript requires substantial support from an experienced English writer with sufficient understanding of the research context.

: Thank you very much for your comments. The manuscript has been re-edited by American Journal Experts, with reference. no 3987-18F0-6D5E-AB8E-72FF . 

In your response to reviewers, please provide the details on how you addressed the comments (what changes were made, where in the text are revised), not just saying those particular points were revised and improved (but what were revised or improved and where). Where data were reanalyzed, explain briefly the new findings and ensure that revisions have been made to reflect the new findings (what were added or removed and where). To most of the comments, particularly those from Reviewer #3, the responses are not sufficient without clear explanations. In your revised manuscript preparation, please have a closer look at the Instructions for Authors and follow them strictly, including word count limit, styles, spacing adjustments, etc.

Abstract

1. The abstract must be condensed as it contains far more than the recommended 350 word count limit.

: The abstract has been revised to 328 words.

2. Line 26, page 1: Please remove ‘sexual problems’ as it is outside of the study’s scope and is not applicable to all target populations under the study (e.g., children).

: Thank you. This term has been deleted.

3. Please use consistent terminologies; e.g., ‘the elderly’ in place of ‘elderly,’ ‘elderly adults,’ ‘elderly individuals,’ ‘older people,’ ‘elderly persons,’ etc. throughout the text.

: Thank you. This terminology has been replaced in the entire text.

4. Line 26, page 1: ‘types of’ is unnecessary.

: Thank you. This phrasing has been improved.

5. Line 30, page 1: The word ‘adults’ is unnecessary.

: Thank you. This term has been removed.

6. Line 1, page 2: A cross-sectional study was conducted among participants who belonged to the Lahu hill tribe and lived in 20 selected villages in Chiang Rai Province, Thailand.

: Thank you. This has been changed.

7. Lines 2-5: A validated questionnaire was used to collect personal information and experiences related to domestic violence in the past year from children aged 5-15 years, women (aged 16-59 years), and elderly aged 60 years and over.

: Thank you. This has been changed. 

8. Line 5, page 2: Please specify whether the ‘logistic regression analyses’ were bivariate, multiple, or both.

: A binary logistic regression was used. The wording has been added.

9. Line 6, page 2: Please remove ‘at the significance level µ=0.05.’

: Thank you. This has been deleted.

10. Results section must be shortened, removing most of the descriptive socio-demographic results.

: Thank you. This has been deleted. Moreover, the abstract has been modified to meet the 350-word limit.

11. The word ‘analysis’ is a countable noun, and its plural form should be ‘analyses.’

: Thank you. However, we changed the word to “in the study”.

12. Line 10, page 2: The use of ‘while’ is incorrect.

: Thank you. This wording has been improved.

13. Line 12, page 2: …children who smoked…

: Thank you. This has been deleted.

14. Line 15, page 2: …a family head…

: Thank you. This has been changed.

15. Lines 14-20, page 2: The large sentence must be restructured as it is difficult to read. It may be broken down into small sentences. Also, when you present the associations between variables, please make the significance level clear, not just say ‘more likely.’

: This has been revised.

Introduction

16. Line 7, page 3: In scientific writing, numbers smaller than 10 should be spelt out.

: Thank you. This has been corrected.

17. Line 19, page 3: …the elderly living in developing countries…

: Thank you. This has been improved.

18. Lines 21-23, page 3: Please revise the sentence to minimize wording redundancies (children, women, the elderly) and improve the referencing. The references 11, 12, and 13 should be moved to join 14 and 15 at the end of the sentence.

: Thank you. This has been improved.

19. Line 24, page 3: Please remove ‘who were’ as it is unnecessary.

: Thank you. This has been improved.

20. Lines 26-27, page 3: “Given people’s economic constraints and substance use, alcohol use has become an integrated factor contributing to domestic violence [18].” This sentence is difficult to understand – how is the first part linked to the latter?

: Thank you. This has been revised and improved.

21. Line 8, page 4: Add ‘,’ before ‘including.’

: Thank you. This has been improved.

Methods

22. Lines 16-17, page 4: The first and second paragraph should be combined as the first one does not make much sense. Consider: “A cross-sectional study was conducted in 20 villages randomly selected out of 216 Lahu villages in Chiang Rai Province, Thailand [21], using a random number generation method.”

: Thank you very much for the help. The paragraphs have been accordingly revised and improved. 

23. Lines 18-19, page 4: Please remove “However, in the study process, only 20 Lahu villages were randomly selected from the list” as does not add any meaning to the text and sounds redundant.

: Thank you. This has been removed.

24. Lines 19-20, page 4: The sentence, “All the Lahu people who were living in the selected 20 villages were invited to participate in the study” is not true as you included only children aged 5-15, women, and the elderly.

: You are correct, and this sentence should not appear in this section. We have modified it and moved it to the next section. 

25. Line 22, page 4: Again, the same sentence was repeated, which should also be removed.

: Thank you for the comment. As we removed the sentence in response to comment no. 24, we need to maintain the sentence in this section. 

26. Lines 24-25, page 4: The sentence “People who were able to provide essential information regarding the study protocol, living selected villages at the date of data collection were eligible for the study” is not readable. Why should the target populations be able to provide essential information regarding the study protocol? Living in the selected villages? The inclusion and exclusion criteria must be clearly stated.

: Thank you. The sentences have been revised.

27. Line 27, page 4: The study sample size…

: Thank you. This has been improved. 

28. Line 1, page 5: “Therefore, the minimum required sample size for the study was 196 participants for each sub-population (children aged 5-15, women aged 16-59, and the elderly aged ³60).” Then you can remove the whole following sentence (However, in this study, three vulnerable groups (children aged 15 years and below, women aged 16-59, and elderly aged 60 year and over) were selected for the study, then, a total of 588 participants were required.”

: Thank you. This has been revised and improved accordingly.

29. Line 7, page 5: Did this study have a conceptual framework? If so, it must be described in the text. If not, this expression about it should be removed.

: Thank you very much for the great comment. This has been removed.

30. Lines 13-15, page 5: The complex sentence needs to be re-structured by breaking it down into a few sentences or using correct punctuations to link the sub-sentences.

: Thank you very much. The sentence has been revised and improved.

31. Lines 17-21, page 5: The long sentence should be rewrite as it is difficult to read and with grammatical errors.

: Thank you very much. This has been improved and shortened. 

32. Lines 25-26, page 5: Why was substance use limited to only cigarette, alcohol, and amphetamine use? Also, how is cigarette smoking related to domestic violence?

: This is because in our literature review, it was found that substances are frequently used by Lahu people. Cigarette smoking is related to amphetamine use, and it was thus included in the study.

33. Lines 27-28, page 5: Are sexual harassment and being forced to have sex considered harm related to sex, not sexual violence?

: Thank you for the comment. We have provided considerable discussion to describe these harms. Basically, sexual violence is a sexual act or attempt to engage in a sexual act by violence or coercion and includes acts to traffic a person or acts against a person directly. This is a clear definition. However, we need to cover only sexual harassment and being forced to have sex, which were specifically found in the Lahu women. Therefore, we decided to use this term. We hope very much that you understand us.

34. Line 30, page 5: The word ‘ignorant’ may have been incorrectly used.

: I am very sorry for the mistake. Actually, the English in this article was improved by American Journal Experts (AJE) before submission. However, I agree with you that there remain many errors. Before resubmission, the article has been re-edited by AJE.

35. Lines 1, page 6: Is ‘domestic abuse’ different from ‘domestic violence?’ It is the first mention of domestic abuse in the paper. Also, please say ‘women aged 16-59’ consistently.

: I am sorry for the confusion. During the literature review, I discovered that scholars use the two terms, domestic abuse and domestic violence, interchangeably. At the time, I wanted to use all words to cover every use and put both words in the paper. However, to be consistent, “domestic abuse” has been deleted. 

36. Line 4, page 6: ‘Cronbach’s alpha.’ Also, state clearly on what is the Cronbach’s alpha for? Domestic violence scale? Across the three participant groups? References to support the scales used in this study must be cited. The use of the scales (coding, recoding, cut-offs used to define the variables…), should be clearly described.

: The Cronbach alpha is used to detect the reliability of questions related to domestic among children, women, and the elderly. I have double checked and out on the numbers and reference in. 

37. Several grammatical mistakes are easily found in the questionnaire development and data collection section. Please revise them carefully. For example, the following three sentences are all incorrect: did you have experienced of kicking out of the house from family member in the past year? Did you have experienced of being forced to drink alcohol, smoke, or use substances in the past year from family member? Did you have experienced of being forced to ask for money or borrow items from others?

: Thank you for the comment. I requested that AJE re-edit the manuscript. I hope that the revised version is acceptable. 

38. Please make ‘statistical/data analysis’ plural (data analyses).

: Thank you. This has been improved.

39. Line 1, page 7: Please specify whether the ‘logistic regression analyses’ were bivariate, multiple, or both. The description of the variable selection for the multiple regression analysis models is hard to understand. The sentences “Some variables were controlled the effect in the model which were determined as the confounder factors for the prediction. In the final model, all significant variables and controlled variables were fitted before making interpretations” are unreadable.

: We used a binary logistic regression with both univariate and multivariate steps.

: The mentioned sentence has been revised.

40. The paragraph on the ethics also needs improvement in writing quality. Did this study need more than one protocol? If so, why? Please correct the wording – e.g., what you need from the participants was their consent, not the form. Please also correct several grammatical errors in the paragraph.

: Thank you very much. I am sorry for the many grammatical errors (even though this version was checked by AJE). The language was improved and revised again by AJE, no. 3987-18F0-6D5E-AB8E-72FF .

Results

All tables and narratives of the tables need to be revised.

41. Lines 19-20, page 7: Please clarify if the 646 was the number of participants recruited or included in the analyses. If the number of the participants recruited was different from that in the analyses, please provide the details of the number of participants excluded with reasons.

: The 646 participants were included in the analyses.

: No participants were excluded from the analyses.

42. Line 22-23, page 7: Please remove ‘While,’ which was incorrectly used, and change ‘elderly persons’ to ‘the elderly.’ Also, please use ‘;’ to separate full sentences, not ‘,’.

: Thank you very much. This has been improved.

43. Lines 21-24, page 7: You don’t need to repeat the number of participants in each group as it was clearly stated in the first sentence.

: Thank you. I agree with you, and this has been improved.

44. Table 1:

• Please do not use ‘.’ in titles.

: This has been removed.

• The use of ‘;’ is also incorrect.

: This has been improved.

• The table’s head is not understandable – Why was the head repeated? Please have a look at other published papers and revise them accordingly

: Thank you for the great comment. I have learned a lot from this comment.

: The table headings have been revised. 

45. Line 2-5, page 9: Please correct all the sentences in this paragraph, which are unreadable and grammatically incorrect.

: All sentences were improved.

46. Table 2:

• The title is incomplete.

: This has been revised and improved.

• The contents of the table are not matched with the table’s head – while the head notes n (%) for Yes and No, one column presents number, the other presents %. Also, it’s unnecessary to present both ‘Yes’ and ‘No’ results. Let readers do the math.

: Thank you very much for the comment. You are correct. This is an important mistake and has been corrected. The column is the n and % of each group.

• While only smoking, alcohol use, and amphetamine use was mentioned in the methods, other types of substances are shown in the table. Please make them consistent. Please also include the last item, ‘Family member who used alcohol’ in the measurements in the methods.

: Thank you for the comment. All types of substance use are mentioned in questionnaire.

: ‘Family member who uses alcohol’ is now mentioned in the Method section.

47. The narrative of Table 3 also needs improvement, correcting the misuse of punctuations typos, and inconsistencies. Also, only one category should be presented in tables for binary variables.

: All the sentences have been revised.

: I am sorry if there is confusion in the presentation. Basically, we want to present the total number of participants and the number that experienced domestic violence. Domestic violence is divided into physical and mental violence. Sexual abuse is presented only for the women.

: We have also resummarized the information. 

48. Line 1, page 14: Please use ‘bivariate logistic regression’ in place of ‘univariate analysis.’ Univariate analyses involve only one variable without a comparison or association and are used to describe participants’ characteristics (e.g., mean, median, %, etc.). Also, ‘smoking behavior’ is not a correct term in this context – you just asked whether they smoked (yes or no), not exploring the way they smoked.

: This section was revised again by AJE.

49. In description of relationships, please specify the direction and significance level of the associations, not just saying the variables are associated.

: Thank you. This has been improved.

50. Since the description of the regression models was not at all clear, the reader cannot understand why only religion and education were controlled for in Table 4. Same for Tables 5 and 6. Also, please use correct terminologies (e.g., multiple logistic regression analysis).

: Thank you. This has been revised and improved.

: After reviewing this point, I completely understand what you mean. The sentences have been revised and improved in all descriptions in Tables 4-6.

51. Lines 2-5, page 14: The two sentences tell the same thing and should be combined. This is also applied to Table 5 and 6.

: Thank you for the comment. All relevant sentences have now been properly combined. 

52. Data in Table 4 tell clearly that the sample sizes were not calculated to represent each study population. As a result, the it was too small for children.

: You are correct. We all have learned a lot from this comment. When we work on the next project, we will carefully consider the sample size; some groups are large, while some groups are small.

: Thank you very much.

53. As commented earlier, please keep only one head for all tables.

: Thank you. All table headings have been revised.

54. The footnotes under Tables 4-6 are not understandable – why were µ values needed to tell a significance level, while p-values were already presented in the tables?

: Thank you. Based on my knowledge, the alpha value is the cut-off point, and we set the statistic to 0.05, while the p-value is obtained from the real data that can be any probability value. So, to determine the significance, we need to compare the alpha value and the p-value before deciding whether to accept or reject the hypothesis. Therefore, please consider this explanation.

Discussion

55. In general, the discussion section requires extensive support from an experience writer. Many statements are not understandable because of the poor quality, many others are too broad and not specifically based on the study findings. The discussions are mostly the comparisons with other studies (A supports B, A is confirmed by B), without context-specific interpretation and linkage to social policy implications.

: Thank you for such a wonderful comment. We have extensively expanded the paragraphs. 

56. Lines 12-14, page 26: The sentence “The WHO [28] reported that 30.0% of domestic violence against women had different prevalence estimates in different communities that were higher than the estimates in the Lahu community in Thailand” is not at all clear.

: Thank you. The sentence has been revised. 

57. Line 16, page 26: Please clarify what ‘clinical populations’ means?

: I am sorry for the unclear interpretation from the original cited paper, which mentioned people attended to in a hospital.

58. Lines 1-12, page 27: The discussion on the relationship between smoking and domestic violence relied so much on personal assumptions that are not well supported by the findings or literature.

: Thank you for the great comment. These references were replaced by the proper references.

59. Lines 13-18, page 27: The paragraph almost unreadable at all due to a poor structure and grammatical mistakes.

: Thank you. The paragraph has been revised.

60. Lines 1-2, page 28: What does ‘…domestic violence was dominated by alcohol use…’ mean? Alcohol use may be a cause of domestic violence, but how it dominates the violence is not understandable.

: You are correct that alcohol use contributes to domestic violence. The sentences have been revised.

61. Limitations are not well presented.

• The sentence does not make sense – the study’s limitations are not determined by the refusal rate.

: Thank you. I agree with you, and the sentences have been deleted.

• It’s not clear how and to what extend data collection by community health volunteers affected the data quality? What measures were taken to address the concerns? Did the volunteers receive any training? Information on data collection training should be clearly provided in the Methods.

: We did not use the community health volunteers to collect the information. We asked them to help in explaining some questions in case the participants did not clearly understand the meaning or the context of the question. This did not occur often because the participants who could not speak Thai were excluded from the study as part of the exclusion criteria. Therefore, we did not train the community participants to collect the data.

• Same for the following sentence. Please discuss clearly what were the methodological issues around the collection of sensitive data?

: Basically, as we know from our more than 10 years of experience in conducting research with Lahu people, domestic violence and other family conflicts are kept secret, and they do not tell these stories in public, particularly to individuals of a different gender. Therefore, all forms of violence are sensitive issues for them. Therefore, we used gender matching during the interview to improve the accuracy of the information.

: We have expanded the explanation in the paragraph.

62. Conclusion also need to be entirely re-written. Please avoid repeating the results or include further discussion. Instead, please summarize key findings and provide relevant recommendations that are well supported by the findings.

: Thank you very much. All of these paragraphs have been rewritten.

References

The reference list does not meet the journal’s requirements. Please rework on it more carefully.

: I am very sorry for the mistake and misunderstanding. I have revised all references to follow the journal guidelines from the following URL: https://journals.plos.org/plosone/s/submission-guidelines#loc-references.

Reviewers' comments:

Reviewer's Responses to Questions

Comments to the Author

1. If the authors have adequately addressed your comments raised in a previous round of review and you feel that this manuscript is now acceptable for publication, you may indicate that here to bypass the “Comments to the Author” section, enter your conflict of interest statement in the “Confidential to Editor” section, and submit your "Accept" recommendation.

Reviewer #2: All comments have been addressed

2. Is the manuscript technically sound, and do the data support the conclusions?

Reviewer #2: Yes

3. Has the statistical analysis been performed appropriately and rigorously? 

Reviewer #2: Yes

4. Have the authors made all data underlying the findings in their manuscript fully available?

Reviewer #2: Yes

5. Is the manuscript presented in an intelligible fashion and written in standard English?

Reviewer #2: Yes

6. Review Comments to the Author

Reviewer #2: Accept. The authors have adequately addressed the comments raised in a previous round of review. This manuscript is now acceptable for publication.

7. PLOS authors have the option to publish the peer review history of their article (what does this mean?). If published, this will include your full peer review and any attached files.

---

## [Editor Report · Decision Letter 2]

2 Mar 2021

Factors associated with domestic violence in the Lahu hill tribe of northern Thailand: A cross-sectional study

PONE-D-20-27497R2

Dear Dr. Apidechkul,

We’re pleased to inform you that your manuscript has been judged scientifically suitable for publication and will be formally accepted for publication once it meets all outstanding technical requirements.

Kind regards,

Siyan Yi, MD, MHSc, PhD

Academic Editor

PLOS ONE
---

## [Editor Report · Acceptance letter]

4 Mar 2021

PONE-D-20-27497R2 

Factors associated with domestic violence in the Lahu hill tribe of northern Thailand: A cross-sectional study 

Dear Dr. Apidechkul:

I'm pleased to inform you that your manuscript has been deemed suitable for publication in PLOS ONE. Congratulations! Your manuscript is now with our production department. 

Kind regards, 

on behalf of

Dr. Siyan Yi 

Academic Editor

PLOS ONE